# Hippocampal and medial prefrontal cortices encode structural task representations following progressive and interleaved training schedules

**Sam C. Berens** [ID]*, **Chris M. Bird**

School of Psychology, University of Sussex, Brighton, United Kingdom

* s.berens@sussex.ac.uk

## Abstract

Memory generalisations may be underpinned by either encoding- or retrieval-based generalisation mechanisms and different training schedules may bias some learners to favour one of these mechanisms over the other. We used a transitive inference task to investigate whether generalisation is influenced by progressive vs randomly interleaved training, and overnight consolidation. On consecutive days, participants learnt pairwise discriminations from two transitive hierarchies before being tested during fMRI. Inference performance was consistently better following progressive training, and for pairs further apart in the transitive hierarchy. BOLD pattern similarity correlated with hierarchical distances in the left hippocampus (HIP) and medial prefrontal cortex (MPFC) following both training schedules. These results are consistent with the use of structural representations that directly encode hierarchical relationships between task features. However, such effects were only observed in the MPFC for recently learnt relationships. Furthermore, the MPFC appeared to maintain structural representations in participants who performed at chance on the inference task. We conclude that humans preferentially employ encoding-based mechanisms to store map-like relational codes that can be used for memory generalisation. These codes are expressed in the HIP and MPFC following both progressive and interleaved training but are not sufficient for accurate inference.

## Author summary

Integrating information across distinct situations allows both humans and non-human animals to solve novel problems. For instance, by observing that topaz is hard enough to scratch quartz, and that quartz is hard enough to scratch gypsum, one can infer that topaz must be harder than gypsum—even if these materials have never been seen together. This type of generalisation (transitive inference) can be achieved by combing different pieces of information either, 1) when an inference is actually needed (retrieval-based generalisation), or 2) when new information is first encountered (encoding-based generalisation). We predicted that the use of these generalisation mechanisms depends on the order in

**Data Availability Statement:** All the data and analysis code required to reproduce each figure and hypothesis test has been uploaded to the Open Science Framework (see https://osf.io/tvk43/).

Links to specific analysis scripts within this repository are provided throughout the Methods section.

**Funding:** This project has received funding from the European Research Council (ERC) under the European Union's Horizon 2020 research and innovation programme, grant number: 819526 to C. M. B. The funders had no role in the study design, data collection and analysis, decision to publish, or preparation of the manuscript.

**Competing interests:** The authors have declared that no competing interests exist.

which information is presented and whether that information was learnt before an overnight rest. Contrary to our predictions, behavioural and neuroimaging analyses of a transitive inference task in humans showed convergent evidence for encoding-based generalisations in all conditions. While these conditions had a large impact on inferential ability, we found that brain regions involved in memory invariably learnt inferred relationships between items that had not been seen together. Strikingly, this appeared to be the case even when participants were unable to make accurate inferences.

## Introduction

Humans are readily able to generalise information learnt in one situation and apply it in another. For example, if we are told that Abuja is generally hotter than Beirut (A>B), and Beirut is hotter than Carlisle (B>C), then we can infer that Abuja is hotter than Carlisle (A>C), despite never having been given that information directly. This particular type of generalisation is known as transitive inference.

The hippocampal system and medial prefrontal cortices (MPFC) have long been implicated in generalising recently learned information for use in new situations. Broadly speaking, contemporary models propose that these generalisations may be supported in two different ways: *1)* retrieval-based models, and *2)* encoding-based models. Despite these opposing views being present in the literature for many decades [1,2], it is unclear which mechanisms are used to support memory generalisation or, indeed, whether one is favoured over the other in particular situations.

Retrieval-based models suggest that the hippocampus encodes pattern-separated representations that express specific relationships between co-presented items [3]. These models argue that generalisation is supported by a recursive neural mechanism that rapidly integrates distinct memories on-the-fly. As such, they predict that the brain only needs to store the originally presented information, since generalisation occurs as and when it is necessary via the retrieval of directly learnt information. Retrieval-based models have received support from both fMRI [4] and behavioural studies [5].

In contrast, encoding-based models suggest that the hippocampal and MPFC systems learn unified representations that directly express inferred structured relationships between task features [6–9]. These 'structural representations' are therefore sufficient to support inference without the need for a specialised inference mechanism. As such, the hallmark of encoding-based models is that the relationships between events have been abstracted and stored, enabling generalisation to occur without the need for online integration. Of course, these knowledge structures may not be created strictly at the point of encoding–it is possible that they emerge after a period of consolidation or after the same information has been experienced several times [10–12].

Consistent with encoding-based models, the hippocampus, entorhinal cortex, and medial prefrontal cortex have been found to encode generalised relationships that were not explicitly trained [6,13–18]. Additionally, the entorhinal cortex and MPFC are known to represent distinct sets of stimuli in similar ways provided that there are common relationships between the stimuli within each set [19–21]. These generalised representations are thought to facilitate inference and knowledge transfer across related tasks, although their relationship to generalisation performance is ambiguous.

Retrieval-based models predict that generalisation performance decreases when inferences require integrating information over more independent memory traces (so-called, negative

transitive slopes; see [3]). However, encoding-based models often predict the opposite relationship, e.g., generalisations are easier when comparing stimuli that are separated by larger distances in an inferred hierarchy (positive transitive slopes). This is because the information required to discriminate stimuli based on their relative positions in an abstract task space becomes increasing salient with larger distances [22]. In support of retrieval-based mechanisms, there is clear evidence of negative transitive slopes when inferences involving distinct episodic memories [5]. Nonetheless, most studies have reported positive transitive slopes consistent with encoding-based mechanisms, particularly in transitive inference paradigms [9,23–26].

Aside from retrieval- and encoding-based models, it has been suggested that above-chance performance on transitive inference tasks can result from stimulus-reward associative learning simply because, during some training procedures, stimuli at the top of the hierarchy tend to be selected more often and so are more commonly associated with reward [27–30]. However, more recent research has shown that these simple associative mechanisms are unable to account for all transitive inference behaviours [9,23,25,26,31–35]. Moreover, participants in the current study were able to perform inferences despite receiving a pattern of reinforcement that was entirely incompatible with stimulus-reward association learning (see https://osf.io/cteg9). As such, we do not consider these models any further in the current study.

Certain training conditions can have a large impact on how information is retained [36] and generalises to new situations [37]. For example, categorisation of previously unseen objects is sometimes improved if exemplars from different categories are presented in an interleaved order, rather than in category-specific blocks, e.g., [38–40]. However, other studies have shown advantages for blocked training schedules, especially when category differences are clearly verbalizable, e.g., [41–43]. Potentially resolving this conflict, interleaving has been shown to aid category generalisation when exemplars are highly similar to one another (both within- and between-categories), whereas blocking may be best when exemplars are relatively distinct [44]. As such, it has been suggested that interleaving emphasises between-category differences, whereas blocking emphasises commonalities amongst exemplars [37,44].

Despite this, it remains unclear whether and how interleaved vs blocked training influences the structure of learnt memory representations, or the generalisation mechanisms that are preferentially employed. Behavioural evidence suggests that blocked training enables the learning of low-dimensional (compressed) stimulus representations that linearly encode task-relevant features [42]. Additionally, recent research shows that training overlapping discriminations in an ordered sequence (e.g., A>B followed by B>C and then C>D, so-called 'chaining') may improve transitive inference by allowing learners to integrate the discriminations into a unified mental model [45,46]. Nevertheless, when this integration takes place, and whether it depends on encoding- or retrieval-based mechanisms has yet to be tested.

We hypothesised that interleaved training would promote the learning of the specific, pattern-separated, pairings and consequently bias the use of retrieval-based inference judgements. This follows the proposal that interleaved training highlights the differences between items. Furthermore, it is consistent with the finding that hippocampal pattern separation prevents interference between overlapping relationships learnt in an interleaved order [47]. In contrast, we hypothesised that presenting related pieces of information in ordered blocks (hereafter referred to as 'progressive training') should facilitate the use of encoding-based inference mechanisms. Specifically, progressive training may enable pattern completion between pairs thereby allowing participants to encode inferred relationships during training [48].

This hypothesis is partly informed by a supplementary analysis demonstrating that, relative to interleaving, progressive training can bias some artificial neural networks to learn task representations that directly encode generalised relationships (see S1 Text). We trained a variety

of multilayer perceptions (MLPs) on a transitive inference task. When the MLPs were constrained to learn low-dimensional (compressed) representations of the discriminations, they tended to encode the relative value of stimuli (i.e., A>B>C). This aided inference performance because the MLPs were not equipped with a retrieval-based generalisation mechanism. Given this result, we predicted that progressive training would facilitate encoding-based inference in humans and speculated that it may confer a similar performance advantage.

As mentioned, many studies have rereported positive transitive slopes that are indicative of encoding-based generalisations. However, it is noteworthy that most of these studies have either employed progressive training schedules [23,26], or provided explicit feedback to inferred discriminations which can confound distance effects with differences in how often stimuli are rewarded [22,24,25]. Here, we directly tested the predictions of retrieval-/encoding-based generalisation mechanisms following both interleaved and progressive training schedules, in a feedback-free inference test.

In addition to the training schedule, we also manipulated whether participants experienced an overnight period of consolidation before being tested on their ability to generalise. Sleep-dependent consolidation has long been implicated in abstracting statistical regularities across separate memories, possibly because it allows distinct event representations to be replayed out-of-order [49,50]. In support of this, many studies have shown that memory generalisation improves following a period of sleep, or even wakeful rest [51–55]. We hypothesised that overnight consolidation would allow pattern-separated memories of task contingencies to be re-encoded as structural memory representations, see [56,57]. We therefore predicted that inferences made on items learnt the previous day would depend more on encoding-based mechanisms than inferences on items learnt immediately prior to scanning.

To test these hypotheses, we analysed the effect of training schedule and overnight consolidation on behavioural and fMRI data collected while human participants performed transitive inferences. We trained a series of 'premise' discriminations via either progressive or interleaved presentations within a reinforcement learning task (see Fig 1). Across consecutive days, 34 participants learnt 2 independent sets of premise discriminations (one set per day), each of which entailed a 1-dimensional transitive hierarchy over 7 visual features (A>B>C>D>E>F>G). Shortly after training on the second day, participants recalled all the premise discriminations and made inferences whilst being scanned. As such, we were able to investigate progressive/interleaved training and inferences based on recent/remote memories in a full factorial design.

We found that progressive training had a large benefit on inference performance in humans. Computational models that captured broad predictions of retrieval- and encoding-based models revealed that our behavioural data are better accounted for by encoding-based mechanisms. Surprisingly, this did not depend on the experimental factors of interest, namely, training schedule and overnight consolidation. Next, we tested neurocognitive predictions of encoding- and retrieval-based models using univariate and multivariate analyses of the imaging data. Retrieval-based models predict the occurrence of BOLD activations that are uniquely associated with on-the-fly generalisation performance. However, we did not observe any effects consistent with this prediction. In contrast, a representational similarity analysis (RSA) supported encoding-based models. Specifically, consistent with multiple encoding-based accounts, we identified RSA effects in the hippocampus and MPFC suggestive of structural representations that expressed inferred relationships across the whole transitive hierarchy. In the MPFC, these effects we only evident for recently learnt contingencies, perhaps suggesting a time-dependent role of this region. We also found that structural representations in the hippocampus and MPFC were associated with different patterns of behavioural performance, perhaps suggestive of different generalisation processes.

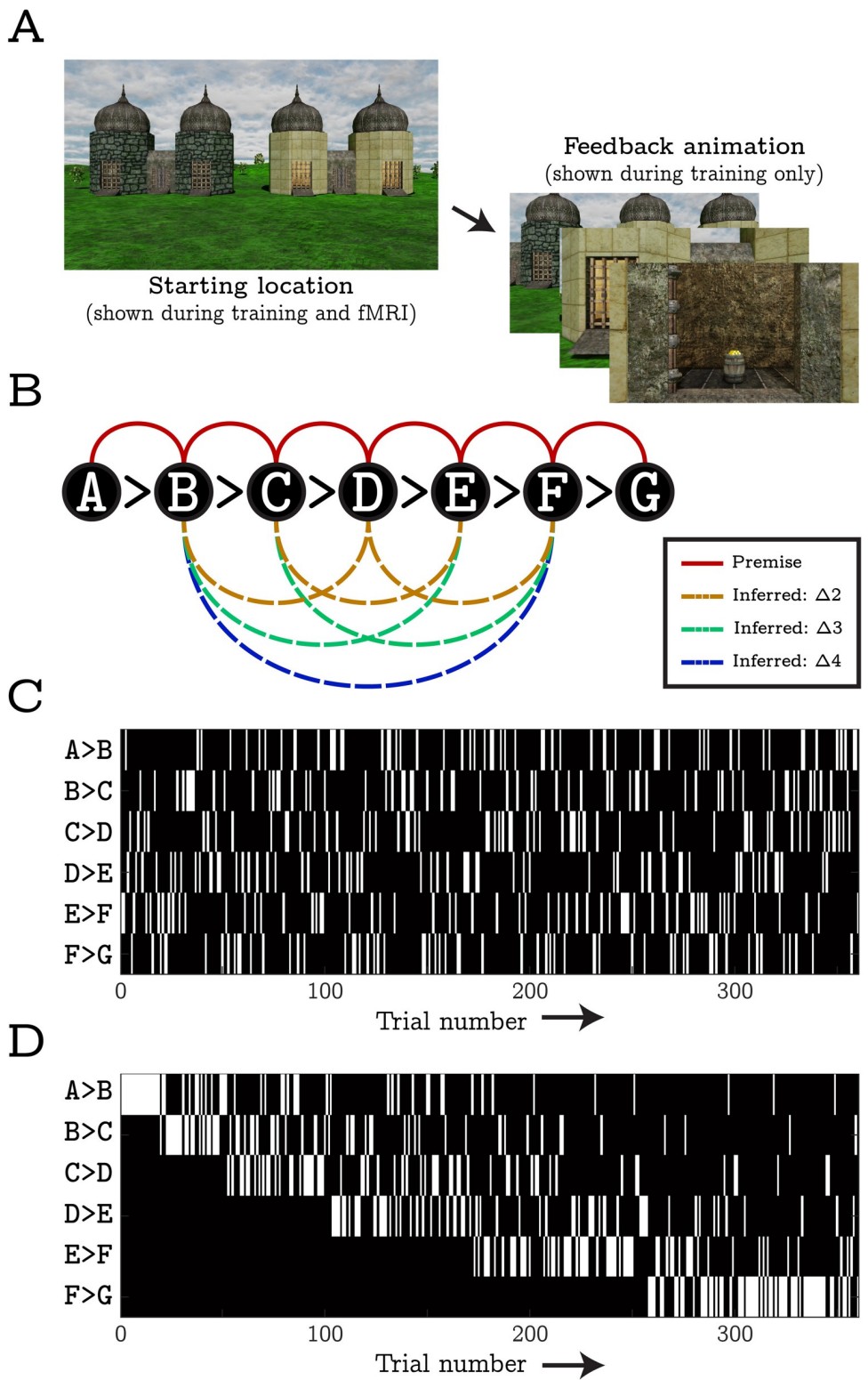

**Fig 1. Illustration of the pre-scanner training and in-scanner behavioural tasks. A)** Both before and during fMRI, participants saw computer-generated images of two buildings with different wall-textures rendered onto their exterior surfaces. One building concealed a pile of virtual gold (reinforcement) and the location of this reward was perfectly determined by the combination of wall-textures shown. In the pre-scanner training phase, participants were tasked with learning the reward contingencies via trial-and-error. A left/right button press was required within 3 seconds of

the start of each trial. Following this, a feedback animation was shown indicating whether the response was correct or not. During the in-scanner task, participants were required to respond to still images of the two buildings, yet no feedback was provided. **B)** A schematic illustration of the reward contingencies trained before scanning (i.e., the premise discriminations, red solid lines) and inferred inside the scanner (i.e., inferred discriminations, dashed lines). Letters denote unique wall textures and the greater than signs indicate the rewarded wall-texture in each premise discrimination. Taken together the 6 premise discriminations implied a 1-dimensional transitive hierarchy. Inferred discriminations did not involve the ends of the hierarchy (i.e., A and G) since such challenges can be solved by retrieving an explicitly trained (featural) contingency (e.g., recalling that A is always rewarded). As such, the set of inferred discriminations included three trials with a 'transitive distance' of Δ2, two trials with a transitive distance of Δ3, and one trial with a transitive distance of Δ4. Note that participants were trained on two independent transitive hierarchies on two separate days: one 24 hours before scanning, one immediately before scanning. While equivalent in structure, the contingencies learnt on each day involved entirely different wall-texture stimuli (counterbalanced across participants) which were never presented in the same trial. **C)** and **D)** On each day of training, premise trials were ordered in one of two ways: interleaved training involved presented all 6 premise discriminations in pseudorandom order such that there was a uniform probability (1/6) of encountering any one discrimination on a particular trial (panel C). In contrast, progressive training involved 6 epochs of different lengths that gradually introduced the discriminations whilst ensuring that, once a discrimination had been introduced, it was presented in all subsequent epochs (panel D).

## Results

### Inference performance

Over two consecutive days, we trained participants to make binary discriminations in a reinforcement learning task (see Fig 1A). Trials presented two buildings that differed in only one respect; the wall textures rendered onto the outside of each building. One building contained a pile of virtual gold (reinforcement) and participants were tasked with learning which wall texture predicted the gold in order to gain as much reinforcement as possible. We explicitly trained 2 sets of discriminations with 6 premise pairs in each; 'A>B', 'B>C' ... 'F>G' (correct responses indicated to the left of the greater-than sign). As such, the contingencies predicting reward implied 2 independent transitive hierarchies (A>B>C>D>E>F>G, Fig 1B).

One set of premise discriminations was trained on each day and training sessions were separated by approximately 24 hours. Prior to the first session, participants were randomly assigned to either an interleaved or progressive training condition which determined the type of training they received on both days. Interleaved training involved presenting all 6 discriminations in a pseudorandom order such that there was a uniform probability (1/6) of encountering any one on a particular trial (see Fig 1C). In contrast, progressive training involved 6 epochs of different lengths that gradually introduced discriminations and ensured that, once introduced, they were presented in all subsequent epochs.

After training on the second day, participants underwent fMRI scanning while recalling all the premise discriminations (from both days) as well as 2 sets of inferred discriminations. As such, the experiment involved 3 main experimental factors: *1)* training method (interleaved vs progressive), *2)* session (recent vs remote), *3)* discrimination type (premise vs inferred). Here, we only report contrasts relating to our a priori hypotheses but a full list of results is available on the Open Science Framework (OSF, https://osf.io/tvk43/).

Fig 2A depicts estimates of performance for the in-scanner task in terms of the probability of a correct response. A mixed-effects logistic regression highlighted similar levels of accuracy for the premise discriminations regardless of training method, session, or their interaction; largest effect: $t(804) = 0.765$, $p = .445$. However, there was a large effect of training method on inference performance with progressive learners outperforming interleaved learners; $t(804) = 5.54$, $p < .001$. This effect was also evident as a main effect of training method (averaged across both premise and inferred trials), $t(804) = 3.83$, $p < .001$, and as an interaction between method and discrimination type; $t(804) = 7.39$, $p < .001$. No main effect of session or a session by method/discrimination type interaction was detected; largest effect: $t(804) = 1.44$, $p = .149$.

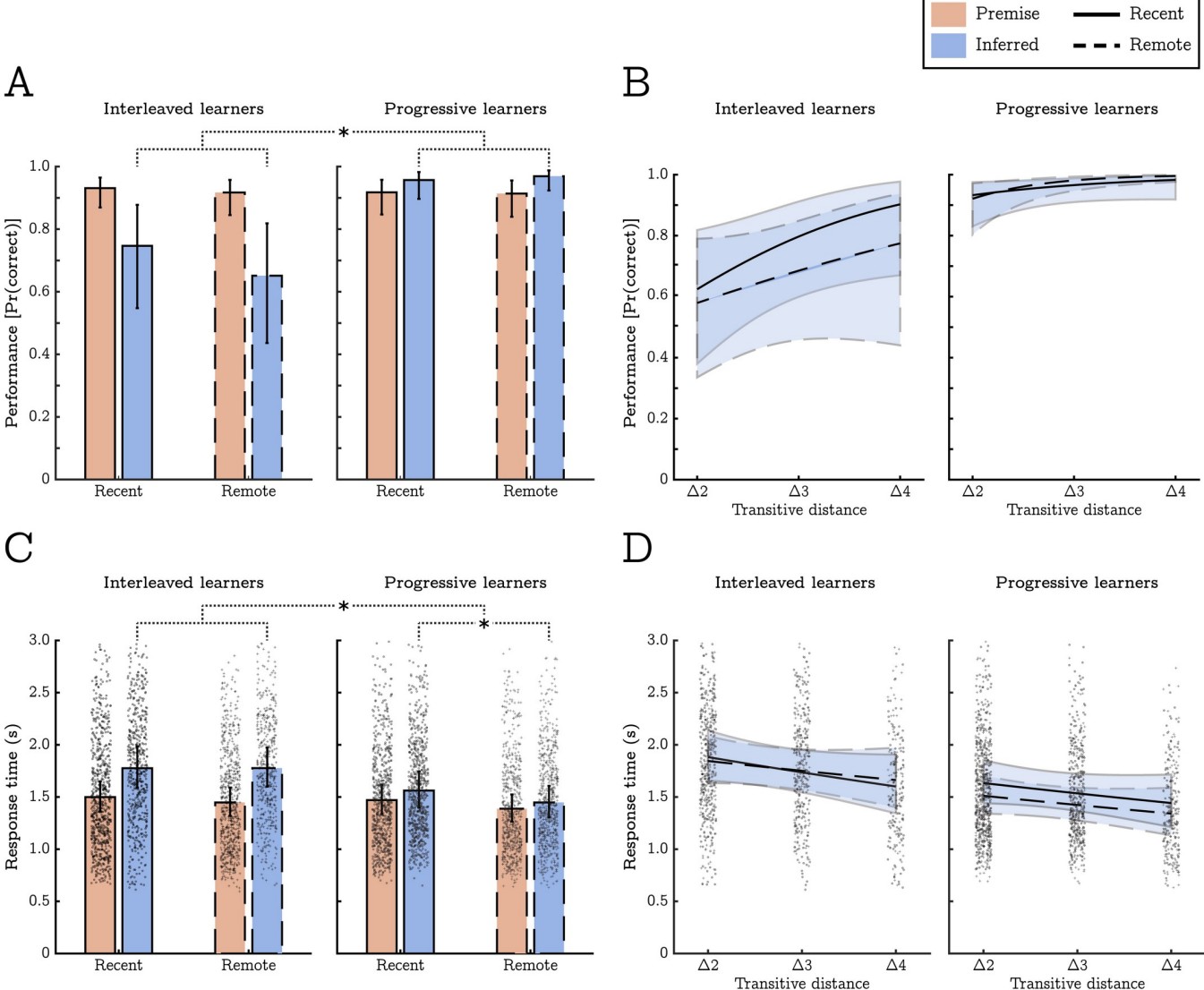

**Fig 2. Humans show better generalisation following progressive training. A)** Estimates of the probability of a correct response, *Pr(correct)*, split by trial type (premise vs inferred) and experimental condition (training method and session). While participants showed comparable levels of performance on the premise discriminations across conditions (red bars), inference performance varied by training method with progressive learners showing much higher levels of accuracy (blue bars). **B)** On inference trials, behavioural performance was positively related to "transitive distance" (the degree of separation between discriminable features along the transitive hierarchy, see Fig 1B). While the correlation between transitive distance and performance was positive in all conditions, the association was most consistent for remote discriminations in the progressive training condition. **C)** Estimates of the mean response time (in seconds, correct responses only) split by trial type and experimental condition (as in panel A). Response times closely mirrored the probability of a correct response but showed an additional effect indicating that participants were faster at responding to remote contingencies (overall). **D)** Response times to inference trails by transitive distance. While not significant, in general, response times decreased as transitive distance increased. Individual data points reflect response times across all trials and participants, and error bars/lines represent 95% confidence intervals.

The logistic regression also examined the effect of 'transitive distance', that is, accuracy differences corresponding to larger or smaller separations between wall textures along the transitive hierarchy (e.g., B>D has a distance of 2, whereas B>F has a distance of 4). We found an overall effect of transitive distance, $t(804) = 2.11$, $p = .035$, indicating that, in general, as the separation between wall textures increased, behavioural accuracy also increased. Additionally, there was a significant 3-way interaction between training method, session and transitive

distance, t(804) = 3.18, p = .002. This suggested that transitive distance was most predictive of performance in the progressive-remote condition, t(804) = 3.07, p = .002, relative to all other conditions which yielded similar distance effects (t-values = 1.99, 1.08, & 1.55, for the inter-leaved-recent, interleaved-remote & progressive-recent conditions respectively, see Fig 2B). However, this interaction does not reflect larger performance increases in the progressive-remote condition as the estimated probability of a correct response was uniformly close to ceil-ing (only increasing from .921 to .996). Instead, it mainly reflects performance becoming more consistently accurate (i.e., lower variances in the binomial probability estimate) with larger distances.

A generalised linear model of response times (correct responses only) produced a comple-mentary pattern of results (see Fig 2C). Specifically, we detected main effects of training method and discrimination type indicating shorter response times from progressive learners and longer response times to inferred discriminations; t(5301) = 2.01, p = .045, and t(5301) = 2.31, p = .021, respectively. These effects were superseded by a training by discrimination type interaction highlighting that longer response times to inferred trials were more pronounced for interleaved learners; t(5301) = 5.17, p < .001. Unlike the accuracy data, this analysis showed a main effect of session indicative of quicker responses to all remote discriminations; t(5301) = 3.26, p = .001. No other significant main effects or interactions were detected.

## Computational models

We predicted that the use of retrieval- and encoding-based generalisation mechanisms would vary by experiment condition. To test this directly, we created two descriptive models based on general principles of retrieval- and encoding-based accounts. Under similar assumptions, each model attempted to predict participants' inference performance from their responses to premise trials. The goodness-of-fit for each model was determined by how well it accounted for the pattern of correct and incorrect responses.

The retrieval-based AND model assumes that correctly inferring a non-trained discrimina-tion (e.g., B>E) involves retrieving all the directly trained response contingencies required to reconstruct the relevant section of the transitive hierarchy (e.g., B>C *and* C>D *and* D>E; so-called mediating contingencies). As such, the AND model captures a general prediction of retrieval-based mechanisms; that inference performance decreases as a function of transitive distance due to the reliance on more independent memory traces [3–5].

In contrast, the encoding-based OR model assumes that participants can access a unified structural representation describing the associative distances between all stimuli. Nonetheless, in order to make a successful inference, knowledge of this associative structure must be evalu-ated alongside the reward contingencies indicating which of the presented stimuli is higher in the reward hierarchy. When making an inference (e.g., B>E), it is therefore sufficient to recall only one of the contingencies indicating which stimulus should be preferred (e.g., B>C), or which stimulus should be avoided (e.g., D>E).

The AND and OR models predict different levels of performance across inference trials (see Methods). We measured the fit of these models against participants' performance data using a cross-entropy cost function and analysed these goodness-of-fit statistics using a generalised linear mixed-effects regression with 3 experimental factors: 1) model type (AND vs OR), 2) training method (interleaved vs progressive), and 3) session (recent vs remote). Fig 3 plots the cross-entropy statistics by all conditions. The mixed-effects regression highlighted main effects of model type, t(128) = 8.45, p < .001, and training method, t(128) = 5.53, p < .001, both of which were qualified by a model type by training method interaction: t(128) = 5.84, p < .001. No other model terms were significant.

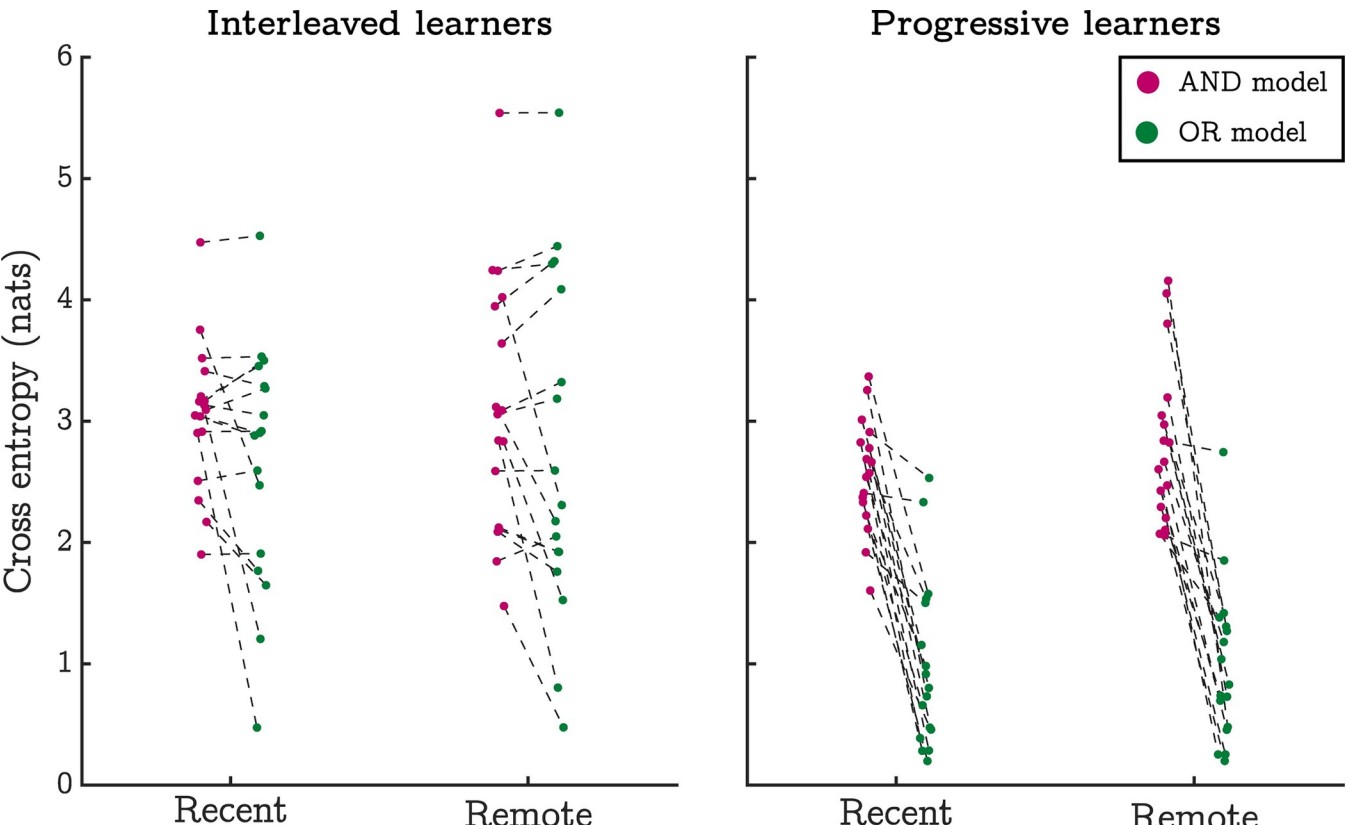

**Fig 3. Behavioural performance is suggestive of encoding-based generalisation mechanisms.** This figure shows goodness-of-fit statistics per participant and condition for two models of inference performance (lower values indicate a better model fit). The AND model implements a general assumption of retrieval-based generalisation mechanisms—that inference requires retrieving multiple independent response contingencies in order to evaluate transitive relationships. In contrast, the OR model realises a general assumption of encoding-based generalisation; specifically, that inferences require the retrieval of a unified structural representation. In all conditions, average goodness-of-fit statistics were lowest for the OR model indicating that it was a better fit to the behavioural data (result qualified by a generalised linear mixed-effects model—see text).

These results indicated that, relative to the AND model, the OR model provided a better fit to the inference data in general, although it was less predictive in interleaved learners. Nonetheless, the OR model was still preferred over the AND model in interleaved learners, t(128) = 2.63, p = .009. This was also evident when we used Spearman rank correlations to compare the number of correct responses to each inferred discrimination with the number of correct responses that would be expected under each model. Specifically, the correspondence between model predictions and the observed data tended to be higher across participants for the OR model in both the progressive and interleaved conditions; t(14) = 7.31, p < .001, and t(16) = 5.38, p < .001 (respectively, statistics derived from bootstrapped paired-samples t-tests, see S1 Table). Contrary to our predictions, these results indicate that inference performance is best accounted for by encoding-based mechanisms in all experimental conditions.

## Univariate BOLD effects

Retrieval-based models of generalisation hold that inferences depend on an online mechanism that retrieves multiple premise contingencies from memory and integrates information between them. As such, we used a set of linear mixed-effects models to test whether BOLD responses were larger on inferred trials than on premise trials and whether this effect was modulated by 5 factors of interest: *1)* transitive distance, *2)* training method (interleaved vs

progressive), *3)* training session (recent vs remote), *4)* inference accuracy, and *5)* the slope relating transitive distance to inference performance (hereafter referred to as the 'transitive slope'). The rationale for this latter factor follows from considering that encoding- and retrieval-based models predict different transitive slopes (being positive and negative, respectively). Given this, the magnitude of the slope can be used to indicate whether BOLD responses more closely adhere to the predictions of one model or the other.

In comparison to the trained discriminations, inference trials evoked lower levels of BOLD in the right hippocampus (specifically, more deactivation relative to the implicit baseline); t(787) = 2.79, p = .005 (S1 Fig). However, this effect was not modulated by training method, t(787) = 0.31, p = .753, or inference accuracy, t(787) = 0.04, p = .965, and so cannot account for variation in inference performance. In contrast, BOLD estimates in the superior MPFC did reflect differences in inference performance. In the left superior MPFC we saw a significant effect of trial type, again indicating more deactivation on inference trials, t(787) = 3.25, p = .001. This was qualified by a 3-way interaction between trial type, training method, and inference accuracy, t(787) = 2.93, p = .003 (Fig 4A and 4B). Similarly, the right superior MPFC produced a significant interaction between trial type and training method t(787) = 2.76, p = .006, (Fig 4C and 4D). Overall, these results indicate that the MPFC produced greater levels of BOLD activity whenever response accuracy was high, regardless of whether participants were responding to premise or inferred discriminations.

In sum, we found no univariate BOLD effects consistent with the use of retrieval-based generalisation mechanisms. While activity in the superior MPFC was associated with behavioural performance, this association was not specific to, or enhanced by novel inferences as would be expected under retrieval-based accounts, see [58,59]. A full list of statistical outputs relating to each ROI is available on the OSF (https://osf.io/sdtyk).

## Representational similarity analyses

We predicted that the training method and the length of the study-test interval would affect how response contingencies were encoded by medial temporal and prefrontal systems. Specifically, we expected that progressive training and longer retention intervals would result in structural representations of the transitive hierarchy and that this would correspond to better inference. To test this, we constructed a series of linear mixed-effects models (LMMs) that aimed to *a)* identify neural signatures of structural memory representations, and *b)* reveal whether they are modulated by each experimental factor (and their interactions).

BOLD responses to each discrimination were first used to estimate representations of individual wall-textures via an ordinary least-squares decomposition (see Methods and Fig 5A). The correlational similarity between wall-texture representations was then analysed in the LMMs to identify 'distance effects' within each hierarchy, i.e., where the similarity between wall-textures from the same transitive chain (i.e., trained on the same day) scaled with transitive distance (e.g., corr[B,C] > corr[B,D] > corr[B,E]). Moreover, the LMMs tested whether such distance effects were modulated by 4 factors of interest: *1)* training method (interleaved vs progressive), *2)* training session (recent vs remote), *3)* inference accuracy, and *4)* transitive slope (as above).

Importantly, the LMMs excluded correlations involving wall textures 'A' and 'G' at the extreme ends of each hierarchy. This is because these stimuli were only presented in premise trials and so their estimated voxel representations may differ from all other representations for trivial reasons. Additionally, each LMM included an extensive set of fixed- and random-effect predictors that controlled for nuisance correlations between co-presented wall textures and correlations resulting from the least-squares decomposition (see Methods).

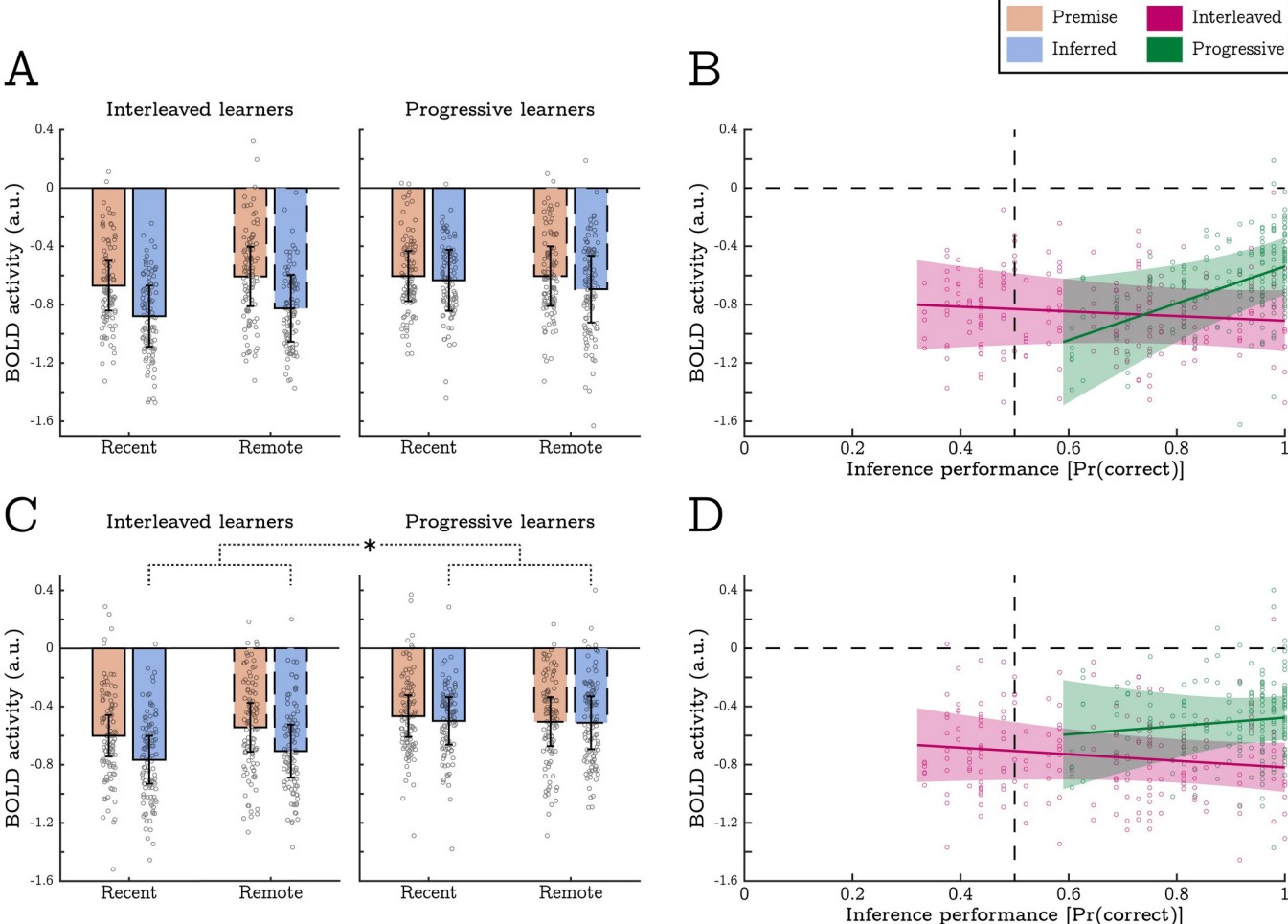

**Fig 4. Inference performance within- and across- experimental conditions is associated with univariate BOLD activity in the superior MPFC.** Panels **A** and **B** show activity in the left superior MPFC. Panels **C** and **D** show activity in the right superior MPFC. Bar charts display mean response amplitudes to all in-scanner discriminations split by trial type (premise vs inferred) and experimental condition (training method and session). Scatter plots display mean response amplitudes to all inference trials (both recent and remote) as a function of inference performance, split by training method (interleaved vs progressive). In the left superior MPFC, a main effect of trial type indicated lower levels of BOLD activity on inference trails (panel A). This was superseded by a significant 3-way interaction indicating larger BOLD responses to inference trials in progressive learners who achieved high levels of inference performance (panel B). The right superior MPFC showed a significant 2-way interaction between trial type and training method. This indicated that BOLD responses in interleaved learners were lower on inference trials (relative to premise trials), but comparable to premise trials in progressive learners (panels C and D). Overall, these data indicate that the MPFC produced greater levels of BOLD activity whenever response accuracy is high. Individual data points indicate discrimination-specific BOLD estimates for each participant and error-bars indicate 95% confidence intervals.

Here, we only report main effects and interactions involving the transitive distance predictor since our hypotheses only concerned these terms. Nonetheless, for completeness, we report all other significant effects in S2 Text and provide a full list of statistical outputs on the OSF (https://osf.io/29x3q). S2 Text also includes an analysis of hierarchical representations that generalised across the transitive chains learnt on each day of training (i.e., across recent and remote conditions). While no significant 'across-hierarchy distance effects' were identified, we detail all other main effects and interactions revealed by this analysis.

In the left hippocampus, we saw a main effect of distance that did not survive our correction for multiple comparisons; $t(652) = 2.61$, $p = .009$. While not reaching our strict criterion for statistical significance, 27 of the 34 participants exhibited the predicted distance effect in this region which is significantly more than would be expected by chance alone ($p < .001$, binomial

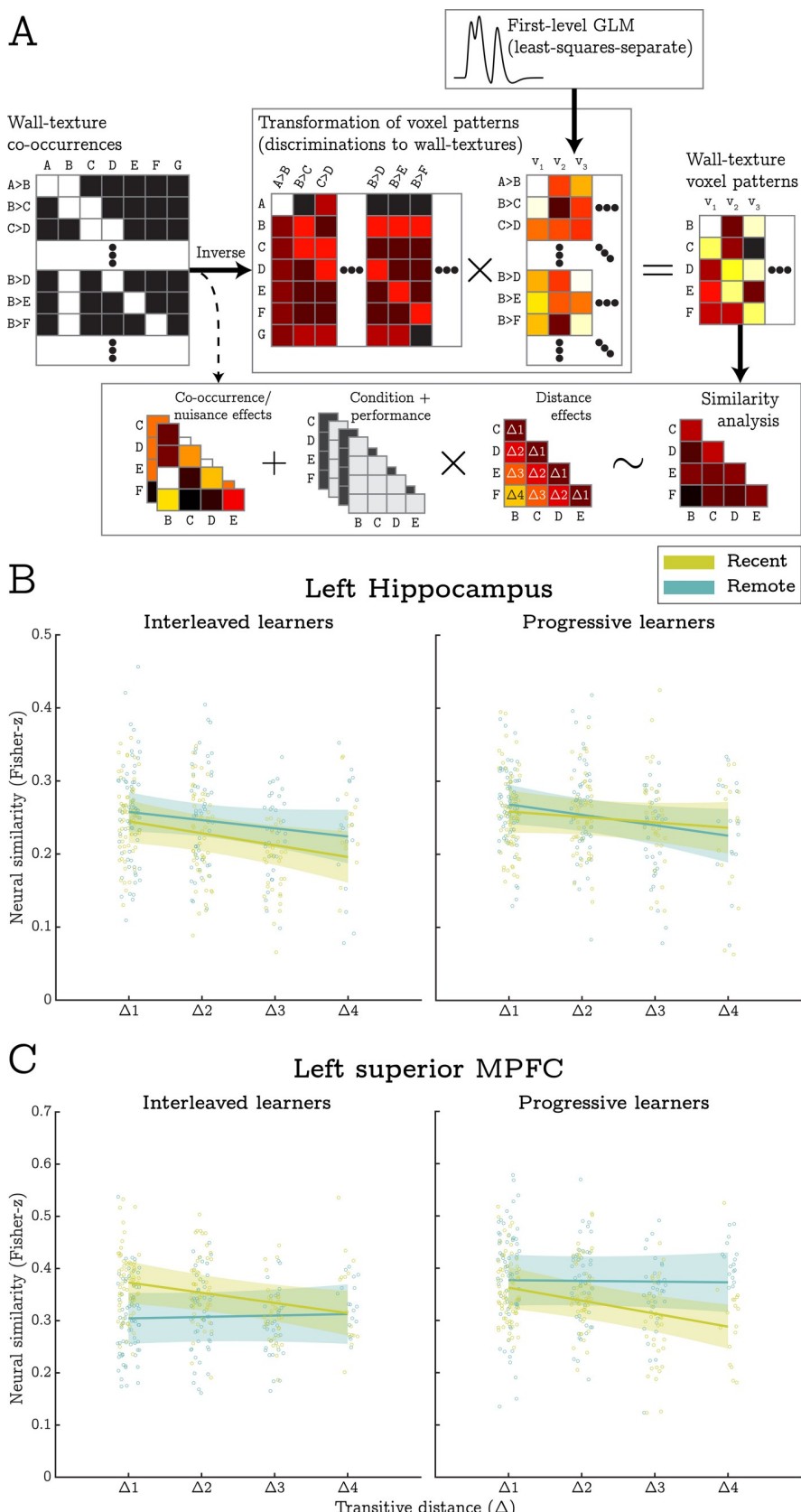

**Fig 5. Methods and results for the RSA. A)** BOLD responses across voxels ($v_1$, $v_2$, etc.) for each in-scanner discrimination (A>B, B>D, etc.) were estimated in a set of $1^{st}$ level models. These were linearly transformed into representations of specific wall-texture stimuli (A, B, C, etc.) via a least-squares decomposition procedure. Subsequently, BOLD similarity between wall-textures was estimated, Fisher-transformed, and entered into a mixed-effects model that implemented the RSA. Nuisance covariates accounted for trivial correlations between co-presented wall-textures and correlations resulting from the decomposition procedure. Effects of interest modelled the influence of condition, behavioural performance, and transitive distance. **B)** In the left hippocampus, transitive distance (i.e., the separation between wall-textures) was negatively correlated with BOLD similarity across all conditions. As such, this region appears to encode a structural representation of the transitive hierarchy that is not modulated by training method (i.e., interleaved vs progressive) or session (recent vs remote). **C)** The left superior MPFC exhibited a distance by session interaction suggesting that structural representations were only expressed for recently learnt contingencies. Individual datapoints indicate pairwise similarity estimates from all participants and shaded error-bars indicate 95% confidence intervals.

test). As such, this effect is consistent with our prediction that representational similarity between wall textures should be inversely related to their hierarchical distance (see Fig 5B). The left hippocampus also produced a distance by transitive slope interaction, t(652) = 3.31, p = .001. Contrary to predictions, this indicated that distance effects were most strongly expressed in participants who had a relatively low (or negative) transitive slope (see Fig 6A). As such, the left hippocampus appears to encode structural task representations most strongly when behavioural performance is less typical of encoding-based generalisation.

In the left superior MPFC, transitive distance was negatively correlated with representational similarity in the recent, but not the remote, conditions; t(650) = 3.59, p < .001, and t(650) = 0.113, p = .910 (respectively). This resulted in a significant interaction between transitive distance and training session suggesting the presence of structural representations for recently learnt stimuli alone, t(650) = 3.00, p = .003 (see Fig 5C). On top of this effect, we saw a 3-way interaction between distance, session and inferential accuracy, t(650) = 4.24, p < .001 (see Fig 6B). This indicated that the strength of structural representations was greatest for participants who did not achieve high levels of inference performance (although note that the distance effect was still significant for the majority of performance scores).

In the right superior MPFC, we also saw a 3-way interaction between distance, session, and accuracy, t(652) = 2.90, p = .004. Again, this was suggestive of structural representations in recent condition, but only when generalisation performance was relatively low, and only for learners in the interleaved training condition. Furthermore, the right superior MPFC produced a distance by transitive slope interaction that was superseded by a 3-way interaction between distance, session, and transitive slope, t(652) = 2.79, p = .006, and t(652) = 2.96, p = .003 (respectively, see Fig 6C). This highlighted that the expected distance effects were only expressed in the remote condition when participants' behavioural data was heavily indicative of encoding-based generalisations. Importantly, this contrasts with the distance effects identified in the left hippocampus which were strongest when participants' behavioural data were *less* indicative of encoding-based generalisation.

Finally, we report a significant 3-way interaction between distance, session, and training method in the left inferior MPFC, t(648) = 2.99, p = .003. This was mainly driven by distance effects in the progressive training condition. Specifically, we observed the predicted negative correlation between distance and pattern similarity in the recent condition (t = 1.95), but the opposing (positive) relationship for progressive learners in the remote condition (t = 2.40). Post-hoc tests indicated that the positive distance effect was principally attributable to high levels of similarity between responses to wall textures 'B' and 'F' (i.e., those with the largest transitive distance in the analysis). While such an effect may reflect how the stimuli were being encoded, this pattern of data was not predicted and so we do not draw any inferences based on this result.

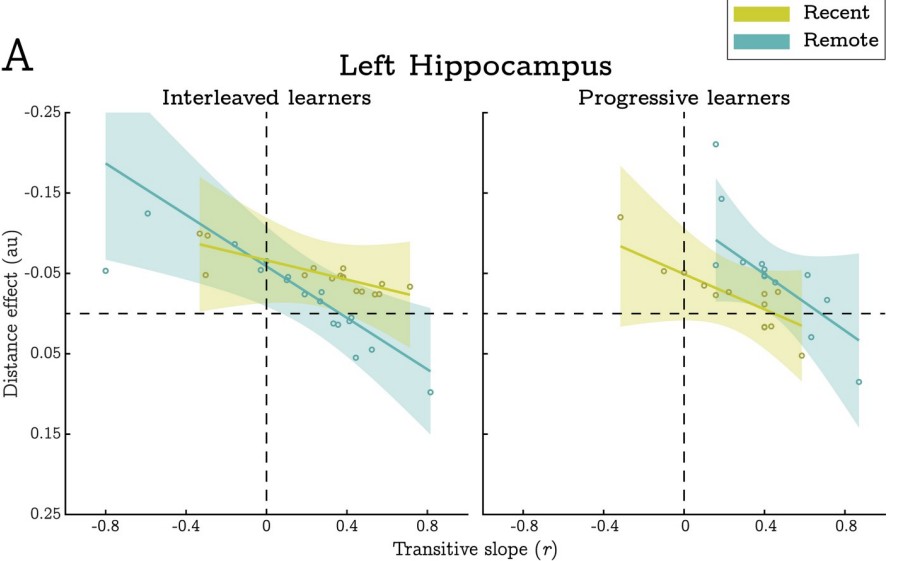

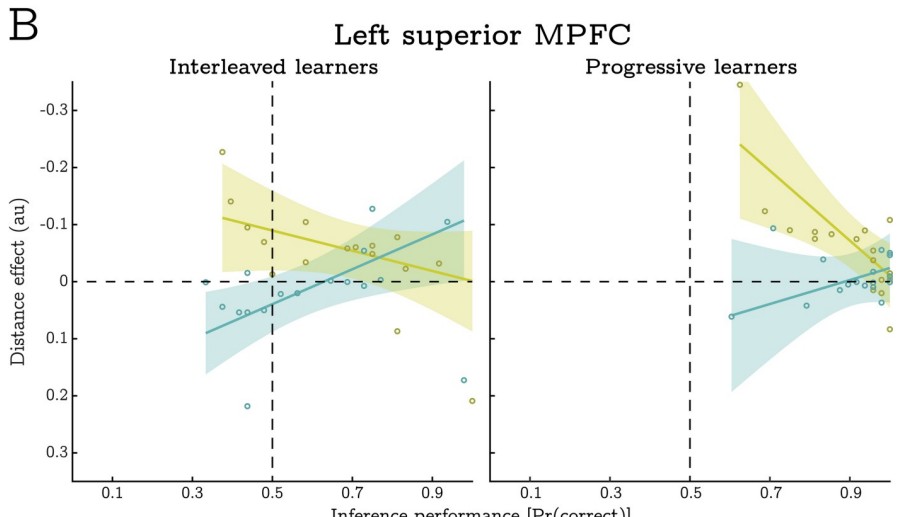

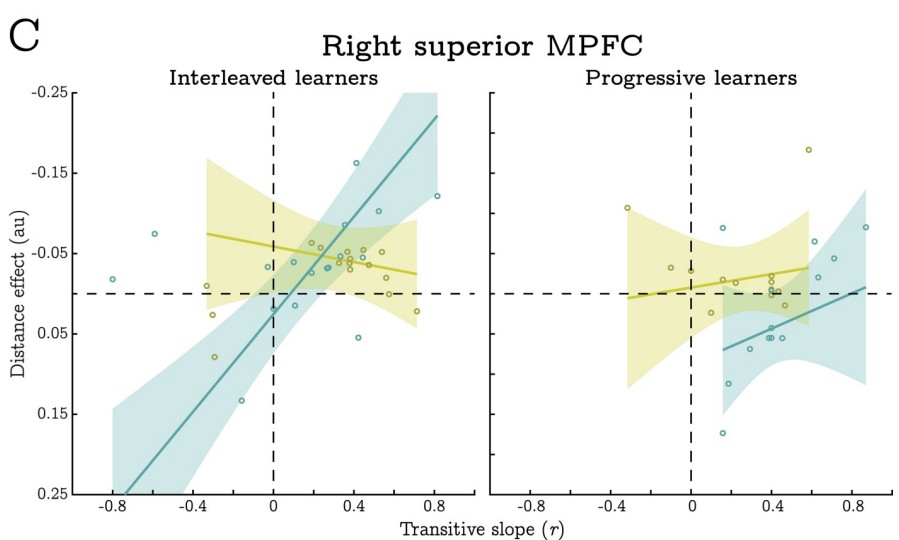

**Fig 6. Associations between measures of behavioural performance and the magnitude of distance effects in the RSA.** Solid trend lines depict the fitted fixed-effect relationship, while shaded error-bars indicate 95% confidence intervals. Note that negative distance effects (plotted above the dashed horizontal) represent the predicted association between transitive distance and BOLD pattern similarity. Individual datapoints depict participant-specific random slopes for each association. **A)** Distance effects in the left hippocampus were strongest when participants produced relatively low transitive slopes (less indicative of encoding-based generalisation). **B)** Distance effects in the left superior MPFC were only significant in the recent condition and were strongest when participants did not achieve high levels of inference performance. **C)** In contrast to the left hippocampus, distance effects in the right superior MPFC were most evident when participants produced relatively large transitive slopes (most indicative of encoding-based generalisation), yet this association was limited to the remote condition. Individual datapoints indicate estimated distance effects per participant and shaded error-bars indicate 95% confidence intervals.

To validate the distance effects reported above, we ran a series of follow-up tests to examine whether they could be attributed to higher levels of BOLD similarity exclusively for stimuli that were shown within the same premise pair (i.e., driven by $\Delta 1$ pairs alone). First, we contrasted similarity estimates between different levels of the distance factor in a random effects analysis. Consistent with our a priori predictions, this invariably revealed significant differences (at uncorrected thresholds) between levels of the distance predictor that did not include $\Delta 1$ pairs. We also re-ran each LMM but excluded similarity estimates between $\Delta 1$ pairs. With the exception of the main effect of distance in the hippocampus, all left lateralised effects reported above survived our correction for multiple comparisons, $t(382) > 2.92$.

Taken together, these findings suggest that the hippocampus contributes to memory generalisations by representing structural memory codes following both interleaved and progressive training. The superior MPFC appears to maintain similar representations, but only for recently learnt material.

## Discussion

We sought to determine whether the use of encoding- and retrieval-based generalisation mechanisms are influenced by two factors: 1) the order in which task contingencies are learnt (i.e., interleaved vs progressive training), and 2) whether there has been a period of overnight consolidation.

Our behavioural analyses demonstrate that progressive training substantially increases generalisation performance compared with randomly interleaving contingencies. This happens despite comparable accuracy in remembering the directly trained (premise) contingencies that generalisations were based upon. However, contrary to our hypotheses, model-based analyses of the behavioural data revealed that encoding-based mechanisms were preferentially used across all experimental conditions. Representational similarity analyses of the fMRI data were also suggestive of the use of encoding-based mechanisms. Here, BOLD pattern similarity in the hippocampal and medial prefrontal cortices correlated with hierarchical distances between stimuli. This implies the presence of map-like structural representations that directly express inferred relationships in an abstract task space. Importantly, these effects were evident following both interleaved and progressive training. It therefore appears that humans have a robust means of learning hierarchical structures regardless of the training method.

In contrast to our results, previous studies have shown that some generalisations are enhanced by interleaved training when compared to blocked schedules. Zhou et al [60] report that both humans and connectionist models of the hippocampal system show elevated levels of memory integration and property generalisation following interleaved training. However, it is notable that our progressive training procedure represents a middle ground between blocking and interleaving. Progressive epochs gradually introduce new discriminations while concurrently testing those that had been shown previously. Unlike fully blocked schedules, this

protects against 'catastrophic interference' [42,61], where new learning results in the forgetting of previously acquired knowledge. It is therefore possible that progressive training offers the best of all worlds–enhanced generalisation across different contexts whilst protecting against catastrophic interference.

To date, state-of-the-art machine learning applications have avoided catastrophic interference by relying on interleaved training [62]. Recently, Kirkpatrick et al [46] introduced an updated loss function for gradient descent that protects previously acquired expertise from catastrophic interference. This cost function incorporates a Fisher information matrix describing how critical each parameter is to maintaining performance on previously trained objectives. It is possible that such protections against interference may allow artificial networks to learn task representations that better support generalisation.

Although progressive training improved inference performance in our study, it appeared to have very little effect on the mechanisms used to make inferential judgements. Analyses of both the behavioural and fMRI data suggested that participants principally used encoding-based mechanisms in all experimental conditions. Moreover, the results of our univariate fMRI analyses did not meet the predictions of retrieval-based mechanisms. The preference for encoding-based strategies that we observed may be due to how well the premise pairs were learnt. Retrieval-based mechanisms are known to explain inference performance when memories have been acquired in a single episode (i.e., one-shot learning; [5]). Given this, our findings are consistent with proposals that encoding-based generalisation mechanisms directly encode abstractions whenever information is frequently rehearsed [49].

As noted, we found evidence that transitive hierarchies were represented in the hippocampus and MPFC in the form of map-like structural codes. This supports various models of memory generalisation that posit representations of physical space can be applied to make inferences in abstract, non-spatial tasks [7,63,64]. Interesting however, the MPFC only appeared to represent structural codes for recently learnt stimuli. Furthermore, while we identified MPFC representations in the majority of participants, they tended to be most strongly expressed in participants who did not achieve high levels of inferential performance. Both of these effects may be attributable to the ease with which structural representations could be accessed and manipulated. It is possible that such operations were more difficult in the recent condition (note the slower response times in this condition), and when participants found inferences more difficult, they tended to activate MPFC representations for longer and/or less efficiently.

Relatedly, BOLD patterns in the MPFC exhibited structural representations even when participants performed at chance level on the inference task (see Fig 6B). It therefore appears that merely having structural representations is not sufficient for good inference performance. Furthermore, while progressive training aided inference performance, we did not detect overall group differences in the strength of structural representations between training conditions. Given this, we speculate that progressive exposure may facilitate the *use* of structural representations during novel inference, rather than the acquisition of structural representations per se. This may explain why some types of generalisation benefit from blocked/progressive exposure, while other forms of generalisation (not dependent on structural codes) do not, e.g., [38–40,60].

Our finding of structural representations in the absence of above-chance inference performance is incompatible with models that propose knowing the relative value of stimuli is all that is needed to build a hierarchical task representation and make novel inferences (e.g., [6], and the MLPs reported in S1 Text). Nevertheless, other models can account for this observation. Neural codes postulated by both the Tolman-Eichenbaum Machine (TEM) and the successor representation (SR) model dissociate the learned values of stimuli from structural

relationships between them [7,63,64]. In the TEM, structural information derived from previous experience is bound to sensory codes in the hippocampus via a fast Hebbian learning rule. However, the ability to use these representations for transitive inference depends on additional path-integration steps that may bottleneck performance. Similarly, SRs can encode the distance between all stimuli in a transitive hierarchy based on knowledge of which stimuli were presented in the same premise pairs. However, in order to support transitive inference, SRs must be combined with a representation encoding the average reward returned by each stimulus.

We also tested whether individual differences in transitive slopes covaried with the strength of structural representations in each ROI. We predicted that participants who produced large, positive transitive slopes (suggestive of encoding-based mechanisms) would most strongly express structural representations in each ROI. This was indeed the case in the right superior MPFC. However, the left hippocampus showed an opposing relationship; a significant distance effect that was strongest for participants who produced negative transitive slopes (see Fig 6). We are unable to fully interpret these results as they were not predicted a priori and because the association between transitive slope and the MPFC distance effect was not consistent across conditions (only being evident in the interleaved, remote condition). However, it is unlikely that there is a simple relationship between the sign and magnitude of each participant's transitive slope, and their use of different generalisation mechanisms.

Aside from a slight decrease in response latencies on both premise and inferred trials, we did not observe any benefit of overnight consolidation on inference performance (in either training condition). This finding is in contrast to a similar test by Ellenbogen et al [54] who found that inference performance increased after a period of sleep. We also did not support our hypothesis that consolidation would bias the use of encoding-based inference mechanisms. Indeed, contrary to this, we found that structural representations in the superior MPFC became less evident for transitive hierarchies that were learnt 24 hours before testing. While it is possible that this effect reflects the consolidation of structural representations outside of the MPFC, or more efficient processing within the region, additional research will be needed to clarify the role of consolidation in memory generalisations.

In summary, we show that progressive training dramatically improves transitive inference and that humans tend to use encoding-based mechanisms to inform inferential judgements based on well-learnt contingencies. Both the hippocampus and MPFC encode structural representations of transitive hierarchies, yet the presence of these representations appears to be insufficient for successful inference. Taken together, these findings provide strong support for encoding-based models that predict map-like structural representations underpin spatial and non-spatial generalisations.

## Methods

### Ethics statement

All participants gave written informed consent on being recruited into the study and were reimbursed for their time. The study was approved by the Brighton and Sussex Medical School's Research Governance and Ethics Committee (project approval code: 15/131/BIR).

### Participants

Right-handed participants were recruited from the University of Sussex, UK. Participants had either normal or corrected-to-normal vision and reported no history of neurological or psychiatric illness. During the study, they were randomly assigned to one of the two between-subject conditions (i.e., the interleaved or progressive training conditions) such that there were an

equal number of useable datasets in each. Data from 5 participants could not be included in the final sample because of problems with fMRI data acquisition (one participant), excessive motion-related artifacts in the imaging data (three participants), and a failure to respond during the in-scanner task (one participant). After these exclusions, the final sample included 34 participants (16 females) with a mean age of 25.9 years (*SD* = 4.60 years).

## Behavioural tasks

**Pre-scanner training.**   We developed a reinforcement learning task designed to train participants on pairwise discriminations before scanning. Two different versions of the task were produced so that each participant could be trained on two occasions; once immediately prior to scanning (recent condition), and once 24 hours before scanning (remote condition).

Unreal Development Kit (Epic Games) was used to generate a number of unique scenes within a first-person virtual environment (see Fig 1A for examples). On each trial, a scene depicted two buildings positioned equidistantly from a start location. One building concealed a pile of virtual gold (reinforcement), yet the only features that predicted the rewarded location were the wall textures rendered onto the towers of each building. Participants were tasked with learning which wall-textures predicted reward in each scene and selecting them in order to gain as much reward as possible.

In total, seven unique wall textures were used in each version of the task. During training, these were combined to generate 6 binary discriminations (e.g., A>B, B>C, etc.) that implied a 1-dimensional transitive hierarchy (A>B>C>D>E>F>G, where each letter denotes a unique wall texture; see Fig 1B). As such, every wall texture could be assigned a scalar value representing its utility in predicting reward. Importantly, each wall texture was rendered onto the left and right buildings an equal number of times to ensure that non-target strategies (e.g., always selecting the building on the left) would not result in above-chance performance.

Fig 1 presents a schematic of the training schedule for participants in either the interleaved or progressive learning conditions. All trials initially depicted the participant at the start location, in front of two buildings, for up to 3 seconds. During this time participants were required to select the building they believed contained the gold via a left/right button press (decision period). Immediately following a response, a 4-second animation was played showing the participant approaching their chosen building and opening its central door to reveal whether or not it contained gold (feedback period). If no response was made within the 3-second response window, a 4-second red fixation cross was shown in place of the feedback video.

For participants in the interleaved learning condition, all discriminations were presented in a pseudorandom order such that there was a uniform probability (1/6) of encountering any one discrimination on any particular trial (see Fig 1C). Given that 'chaining' overlapping trials in an ordered sequence (e.g., 'B>C' followed by 'C>D') may facilitate encoding-based generalisations [45,46], it is noteworthy that our interleaved training schedule presented participants is relatively few chained sequences. Specifically, out of 360 training trails, there were an average of 75.12 chains with a length of two, 10.23 chains with a length of three, and a negligible number of chains (1.51) with a length of four or more discriminations.

For participants in the progressive learning condition, the task was composed of 6 sequentially presented epochs of different lengths which gradually introduced each discrimination one-by-one. The first epoch exclusively trained the discrimination at the top of the transitive hierarchy (A>B) across 17 trials. The second epoch involved an additional 14 trials of the A>B discrimination but also introduced the next-highest discrimination (B>C) across 20 trials (~59%). This pattern continued down the hierarchy such that, after a discrimination had been introduced, the number of times it was tested in subsequent epochs linearly decreased

but remained above zero so that all discriminations were tested in the final epoch (see Fig 1D). Full details of this training procedure are provided on the OSF (https://osf.io/uzyb7/).

Regardless of the training condition that participants were assigned to, all pairwise discriminations were tested 60 times each by the end of the training procedure (i.e., 360 trials in total, ~37 minutes). Before the first training session, participants were briefed on the experimental procedure and told that the wall textures were the only features that predicted reward. These instructions specified that each wall texture should be considered as a single separate 'pattern', and that may either conceal the reward or not, depending on the other wall texture presented within the scene. They were not given any other details regarding the number or type of discriminations.

**In-scanner task.**   Following the second training session, participants were tested on the 6 directly trained (premise) discriminations, and a set of 6 transitive inferences (e.g., B>D), whilst being scanned (see Fig 1B). This tapped knowledge acquired during both of the preceding training sessions. Note that the inferred discriminations did not involve wall-texture stimuli from the ends of each hierarchy (i.e., A and G). This is because discriminations involving these terminal stimuli may be made by applying simple feature-based response policies (i.e., "Always select A", "Always avoid G"), without the need to use a generalised value function.

Similar to the training task, all in-scanner trials initially depicted the participant at a start location in front of two buildings. Participants were instructed to select the building that they believed contained virtual gold based on what they had learned during training. Guesses were strongly encouraged if the participant was not confident. Unlike the previous training sessions, the image of the start location persisted on-screen throughout the 3-second response window regardless of when/whether a response was made. Importantly, no feedback videos were shown during the in-scanner task meaning that participants could not (re-)learn the contingencies via external feedback. Following the response window, a fixation cross was displayed centrally for 3.5 seconds before the next trial commenced.

The in-scanner task tested each premise/inferred discrimination 8 times (the higher value wall-texture appeared on the left-hand building in exactly 50% of trials). As such, the task involved a total of 192 trials: 2 trial types (premise vs inferred) x 2 training sessions (recent vs remote) x 6 unique discriminations x 8 repetitions. Additionally, we included 16 null events (lasting 6.5 seconds each) in order to facilitate the estimation of a resting baseline. All trials were presented in a single run and progressed in a pseudorandom order that was determined by an optimization procedure to enable maximally efficient decoding of trial-specific BOLD responses (https://osf.io/eczjf/).

**Refresher task.**   As noted, participants were trained on 2 independent sets of premise discriminations in pre-scanner training sessions that occurred approximately 24 hours apart. The wall-textures used in each session were counterbalanced across participants. Just before entering the scanner, participants practised each of the directly trained discriminations in a short refresher task. This ran identically to the training tasks but only included 12 trials of each discrimination (lasting approximately 15 minutes). The refresher was not intended to act as an additional training phase but served to remind participants of the appearance of all wall textures so that they were easily identifiable.

## Analysis of in-scanner performance

We used a generalised-linear mixed-effects model (GLMM) to characterise the pattern of correct vs incorrect responses during the in-scanner task. Specifically, this tested the relationship between response accuracy and 3 binary-coded fixed-effect predictors: *1)* trial type (premise vs inferred), *2)* training method (interleaved vs progressive), and *3)* training session (recent vs

remote). Additionally, a continuous (mean-centred) fixed-effect predictor accounted for the effect of transitive distance on inference trials. All possible interactions between these variables were included meaning that the model consisted of 12 fixed-effects coefficients in total (including the intercept term). We also included random intercepts and slopes for each within-subject variable (grouped by participant), and random intercepts for each unique wall-texture discrimination (to account for any stimulus specific effects). Covariance components between random effects were fully estimated from the data.

The outcome variable was the number of correct responses to the 8 repeated trials for each in-scanner discrimination. This outcome was modelled as a binomial process such that parameter estimates encoded the probability of a correct response on a single trial, $Pr(correct)$. To avoid any biases resulting from failures to respond (1.81% of trials on average), we resampled missing responses as random guesses with a 50% probability of success. The model used a logit link-function and was estimated via maximum pseudo-likelihood using the Statistics and Machine Learning toolbox in MATLAB R2020a (The MathWorks Inc.).

In addition to the model of response accuracy, we estimated a similar GLMM that characterised behavioural patterns in response latencies (correct trials only). This GLMM used exactly the same fixed- and random-effect predictors as above. Response times were modelled using a log link-function and the distribution of observations was parameterised by the gamma distribution. As before, the model was fit via maximum pseudo-likelihood in MATLAB.

## Computational models of inference performance

We predicted that inference performance would vary by experimental condition due to differences in the way inferences were made, but not because of any differences in performance for the directly trained discriminations. To test this, we produced two competing models of the behavioural data referred to as the AND and OR models. Both of these attempted to predict participants' inference performance given responses to the directly trained discriminations alone.

The AND model assumes that correctly inferring a non-trained discrimination (e.g., 'B>E') involves retrieving all the directly learnt response contingencies required to reconstruct the relevant transitive hierarchy (e.g., 'B>C' and 'C>D' and 'D>E'). We refer to these directly trained discriminations as "mediating contingencies". As such, this model captures a common assumption of retrieval-based models of generalisation.

In contrast, the OR model assumes that participants have access to a unified structural representation describing the associative distances between all stimuli. Nonetheless, in order to make a successful inference, knowledge of this associative structure must be evaluated alongside the reward contingencies indicating which of the presented stimuli is higher in the reward hierarchy. When making an inference (e.g., B>E), it is therefore sufficient to recall only one of the contingencies indicating which stimulus should be preferred (e.g., B>C), or which stimulus should be avoided (e.g., D>E).

Note, these models are not intended to be process models that describe how humans solve the task at the algorithmic level. They are merely intended to describe the data and test whether the behaviour in each condition better accords with general predictions of encoding and retrieval models.

To formalise both models, we first computed a likelihood function describing plausible values for the probability of correctly retrieving each premise discrimination ($r_p$, where the index $p$ denotes a specific premise discrimination). To do this we assume the probability of observing $k_p$ correct responses, to the $n = 8$ test trials, depends on a joint binomial process involving $r_p$

and, if retrieval is not successful, a random guess that yields a correct response with a probability of 0.5:

$$\Pr\left(k_p | n, r_p\right) = \sum_{m=0}^{k_p} \frac{n!}{m!(k_p - m)!(n - k_p)!} r_p^m (1 - r_p)^{n-m} (1/2)^{n-m} \tag{1}$$

From this, the likelihood function for the parameter $r_p$ (denoted $L(r_p | k_p, n)$) is given by dividing out a normalising constant, $c(k_p | n)$, computed by numerical integration:

$$L\left(r_p | k_p, n\right) = \frac{\Pr(k_p | n, r_p)}{c(k_p | n)} \tag{2}$$

Where:

$$c(k_p | n) = \int_0^1 \Pr(k_p | n, r) d\,r \tag{3}$$

S2A Fig displays the likelihood function for $r_p$ under different values of $k_p$. Based on these likelihoods, we then sampled random values of $r_p$ for each premise discrimination that mediated the generalisation trails. To do this, we used an inverse transform sampling method where a value of $r_p$ was selected such that the cumulative likelihood up to that value (i.e., $\int_0^{r_p} L(x | k_p, n) dx$) was equal to a unique, uniformly distributed random number in the range [0, 1] (see S2B Fig).

As noted, the AND model assumes that a non-trained discrimination depends on successfully retrieving all the reward contingencies that span the transitive hierarchy between presented stimuli. We denote the set of sampled $r_p$ values related to these mediating contingencies $A_i$, where the index $i$ denotes a specific non-trained discrimination, and the number of elements in $A_i$ is equal to the transitive distance. Given the sampled values in $A_i$, we therefore computed the probability this for each non-trained discrimination (denoted $g_i^{and}$):

$$g_i^{and} = \kappa \prod_{r_p \in A_i} r_p \tag{4}$$

The constant term $\kappa$ is a scalar value in the range [0, 1] that determines the probability of engaging in memory-guided generalisations rather than simply guessing. This parameter was fit to the inference data by a bounded nonlinear optimiser ("fmincon", MATLAB Optimization Toolbox, R2020a). Specifically, given the sampled values in $A_i$, the optimiser was tasked with finding a single value of $\kappa$ across all inference trials from a particular participant/condition that minimised a cross-entropy term ($H$) relating model predictions to the observed inference data (see https://osf.io/a6w9t). $H$ was based on Eq 7 and describes the mean log-probability of observing $k_i$ correct responses to all inference trials.

The OR model assumes that performance on the non-trained discriminations depends on successfully retrieving either of the reward contingencies indicating which stimulus should be preferred or avoided. We denote the set of sampled $r_p$ values related to these two contingencies $O_i$, and computed the probability of successful inference under the OR model ($g_i^{or}$) as follows:

$$g_i^{or} = \kappa(1 - \prod_{r_p \in O_i} (1 - r_p)) \tag{5}$$

Note that the value of $\kappa$ was estimated as above, but independently for each model. We then computed model-derived probabilities for the number of correct inference responses $k_i$ to the

$n = 8$ inference trials (similar to *Eq 1*):

$$\Pr(k_i|n, g_i) = \sum\nolimits_{m=0}^{k_i} \frac{n!}{m!(k_i - m)!(n - k_i)!} g_i^m (1 - g_i)^{n-m} (1/2)^{n-m} \qquad (6)$$

In order to estimate the expected distribution of $Pr(k_i|n, g_i)$ for each type of inference, we repeatedly sampled sets of $R_i$ over 10000 iterations using the aforementioned likelihood functions (*Eq 2*). The cross-entropy $H$ of each model was then taken as the mean negative log probability over all $I$ inferences in a particular condition, from a particular participant:

$$H = -\frac{1}{I} \sum\nolimits_{i=1}^{I} \log(\Pr(k_i|n, g_i)) \qquad (7)$$

To analyse condition-dependent differences in the cross-entropy statistics, we entered them into a GLMM with 3 binary-coded fixed effect predictors: *1)* inference model (AND vs OR), *2)* training method (interleaved vs progressive), and *3)* training session (recent vs remote). All possible interactions between these predictors were also included. The GLMM further contained random intercepts and slopes of each fixed effect (grouped by participant), with a covariance pattern that was fully estimated from the data. Cross-entropy was modelled using a log link-function and the distribution of observations was parameterised by the gamma distribution. The model was fitted via maximum pseudo-likelihood in MATLAB.

Although the gradient of transitive slopes may also be used to differentiate encoding vs retrieval-based mechanisms [5,26], the approach outlined above explicitly accounts for differences in premise trial performance that can otherwise confound the analysis. For instance, discriminations between stimuli separated by a larger transitive distance are more likely to involve reward contingencies that can be remembered either better or worse than most others. This may have non-linear effects on inferential accuracy thereby obscuring, or even reversing, the direction of transitive slopes. Our computational models overcome this problem by explicitly accounting for the profile of premise trial performance.

## MRI acquisition

All functional and structural volumes were acquired on a 1.5 Tesla Siemens Avanto scanner equipped with a 32-channel phased-array head coil. T2*-weighted scans were acquired with echo-planar imaging (EPI), 34 axial slices (approximately 30° to AC-PC line; interleaved) and the following parameters: repetition time = 2520 ms, echo time = 43 ms, flip angle = 90°, slice thickness = 3 mm, inter-slice gap = 0.6 mm, in-plane resolution = 3 × 3 mm. The number of volumes acquired during the in-scanner task was 537. To allow for T1 equilibrium, the first 3 EPI volumes were acquired prior to the task starting and then discarded. Subsequently, a field map was captured to allow the correction of geometric distortions caused by field inhomogeneity (see the MRI pre-processing section below). Finally, for purposes of co-registration and image normalization, a whole-brain T1-weighted structural scan was acquired with a $1mm^3$ resolution using a magnetization-prepared rapid gradient echo pulse sequence.

## MRI pre-processing

Image pre-processing was performed in SPM12 (www.fil.ion.ucl.ac.uk/spm). This involved spatially realigning all EPI volumes to the first image in the time series. At the same time, images were corrected for geometric distortions caused by field inhomogeneities (as well as the interaction between motion and such distortions) using the Realign and Unwarp algorithms in SPM [65,66]. All BOLD effects of interest were derived from a set of first-level general linear models (GLM) of the unsmoothed EPI data in native space. Here, we estimated

univariate responses to the 24 discriminations (i.e., 6 premise + 6 inferred, from each day) using the least-squares-separate method [67]. To do this, a unique GLM was constructed for each discrimination such that one event regressor modelled the effect of that discrimination while a second regressor accounted for all other discriminations. As such, one beta estimate from each model encoded the BOLD response for a particular discrimination. These models also included the following nuisance regressors: 6 affine motion parameters, their first-order derivatives, squared values of the motion parameters and derivatives, and a Fourier basis set implementing a 1/128 Hz high-pass filter.

For the analysis of univariate BOLD activity, the 24 beta estimates related to each discrimination were averaged within regions of interest and entered into a linear mixed-effects regression model (see 'Analysis of univariate BOLD' below). For the RSA, these beta estimates were linearly decomposed into voxel-wise representations of each wall texture in the reinforcement learning task (n = 7 per transitive chain). This decomposition involved multiplying the 24 beta values from a given voxel with a 15*24 transformation matrix that encoded the occurrence of each wall texture across discriminations (see Fig 5A). The first 7 outputs of this transformation related to the recently learnt wall textures, the second 7 outputs related to remotely learnt wall textures, and the final output encoded overall BOLD differences between premise and inferred trials (a nuisance term that was not included in any further analysis). Importantly, BOLD representations for wall textures 'A' and 'G' may have trivially differed from all other patterns since these stimuli were only presented in premise trails and so were only shown alongside one other wall texture ('B' and 'F', respectively). As such, similarity scores involving the 'A' and 'G' patterns were excluded from the RSA leaving only scores related to 'B', 'C', 'D', 'E', and 'F'.

## Regions of interest

Numerous studies have implicated the hippocampus, entorhinal cortex, and medial prefrontal cortices in memory generalisations [4,7,19,10–17]. As such, our a priori ROIs comprised 8 binary masks that covered all these areas in native space (separately in each hemisphere). This was done by transforming group-level masks in MNI space using the inverse warp utility in SPM12. For the hippocampus, we used an MNI mask provided by Ritchey et al [68]. The entorhinal masks were derived from the maximum probability tissue labels provided by Neuromorphometrics Inc. Finally, 4 separate masks corresponding to the left and right inferior and superior MPFC were defined from a parcellation that divided the cortex into 100 clusters based on 17 resting-state networks identified by Schaefer et al [69]. Normalised group averages of each ROI used in our main analyses are shown in S3 Fig and are available at https://osf.io/tvk43/.

Notably, the MPFC ROIs that we selected for a priori analyses were relatively large compared to the hippocampal and entorhinal ROIs. As such, we provide supplementary analyses of the MPFC based on a finer, 400 cluster, parcellation of the Schaefer et al networks (see S3 Text).

## Analysis of univariate BOLD

Univariate BOLD effects were investigated within a set of linear mixed-effects models (LMMs). These characterised condition-dependent differences in ROI-averaged beta estimates that derived from a first-level GLM of the in-scanner task (see 'MRI pre-processing' above). The LMMs included 3 binary-coded fixed-effect predictor variables: *1)* trial type (premise vs inferred), *2)* training method (interleaved vs progressive), and *3)* training session (recent vs remote). Additionally, 3 mean-centred continuous fixed-effects were included: *i)* inference

accuracy (averaged across discriminations, per participant, per session), *ii)* 'transitive slope' (the simple correlation between transitive distance and accuracy, per participant, per session), and *iii)* transitive distance per se (applied to inference trials only). All interactions between these variables were also included (excluding interactions between the continuous predictors) meaning that the model consisted of 28 fixed-effects coefficients in total (including the intercept term). We also included random intercepts and slopes for each within-subject fixed-effect (group by participant), as well as random intercepts for each unique wall-texture discrimination (both grouped and ungrouped by participant). Covariance components between random effects were fully estimated from the data. The model used an identity link-function and was estimated via maximum likelihood in MATLAB.

### Representational similarity analysis

Condition-dependent differences in the similarity between wall-texture representations were also investigated using LMMs. To generate these models, we first estimated BOLD similarity in each ROI by producing a pattern-by-pattern correlation matrix from the decomposed wall-texture representations (including wall textures 'B' to 'F' only, see 'MRI pre-processing'). The resulting correlation coefficients were then Fisher-transformed before being entered into each LMM as an outcome variable. These models were structured to predict the Fisher-transformed similarity scores as a function of various predictors of interest. As above, covariance components between random effects were fully estimated from the data. The models used an identity link-function and were estimated via maximum likelihood in MATLAB.

Critically, the temporal structure of the in-scanner task and least-squared decomposition procedure introduced nuisance correlations between wall texture representations. To account for these, we derived two predictor variables of no-interest and used them to model nuisance effects in each LMM described below. The first predictor accounted for trivial correlations resulting from shared sources of noise across co-presented wall textures. This was taken as the Fisher-transformed correlation between binary vectors encoding whether each pair of wall textures were presented in same the in-scanner trails. Across analyses, this predictor invariably accounted for a significant amount of variance in the similarity scores, $r \in [0.105, 0.277]$.

The second predictor of no-interest modelled nuisance correlations attributable to the temporal proximity of trials during the in-scanner task and the pattern decomposition procedure itself. To estimate these correlations, we simulated independent, normally distributed voxel patterns for all wall textures across a large number of iterations, mixed them together in accordance with the trial timings for each subject, and re-estimated the voxel patterns using the least-squares-separate decomposition procedure outlined above. The predictor of no-interest was then taken as the mean Fisher-transformed correlation between simulated wall-textures across iterations. Using this procedure, we aimed to model the effect of fMRI repetition suppression by parametrically modulating the simulated BOLD responses such that repeated presentations evoked an attenuated response. This adjustment was based on Fritsche et al [70] who report that repetition suppression effects in the parahippocampal cortex yield a BOLD attenuation of approximately 23% following a 100ms delay, and 10% following a 1-second delay. Given this, we applied repetition suppression effects assuming an exponential recovery from adaptation over time. Our simulations showed that the presence of such effects did not notably bias BOLD pattern recovery. Furthermore, the temporal signal-to-noise ratio of the fMRI signal had little effect on the correlational structure of the recovered BOLD patterns. Nonetheless, the simulations did reveal some minor nuisance correlations that tended to account for a significant amount of variance in the pattern similarity scores, $r \in [0.001, 0.183]$.

## Within-hierarchy RSA

The first set of similarity analyses tested for differences between wall-texture representations from the same transitive hierarchy. Similar to the models described previously, these LMMs included 5 fixed-effect predictors of interest: *1)* training method, *2)* training session, *3)* transitive distance, *4)* inference accuracy, and *5)* transitive slope. All interactions between these variables were also included (excluding interactions between inference accuracy and transitive slope). The LMMs also included random intercepts and slopes for each effect derived from a repeated measure variable. Finally, the models comprised an extensive set of random intercepts and slopes (grouped by participant) that accounted for all dependencies between pattern correlations (see https://osf.io/jwaek).

## Across-hierarchy RSA

The second set of similarity analyses tested for differences between wall-texture representations from different transitive hierarchies (i.e., those learnt in different training sessions). These LMMs included 4 fixed-effect predictors of interest: *1)* training method, *2)* transitive distance, *3)* inference accuracy, and *4)* transitive slope. Note that the effect of training session was not included as did not apply when examining the similarity between representations learnt in different sessions. As before, the effect of transitive distance accounted for comparisons between wall-textures at different levels of the hierarchy. However, in this set of models, the distance predictor included an additional level ($\Delta 0$), corresponding to comparisons between wall-textures at the same hierarchical level. The across-hierarchy LMMs included the same nuisance variables and random-effects as in the within-hierarchy RSA (https://osf.io/cjv7h).

## Statistical validation and inference

To ensure that each linear mixed-effects regression model was not unduly influenced by outlying data points, we systematically excluded observations that produced unexpectedly large residual values above or below model estimates. The threshold for excluding data points was based on the number of observations in each model rather than a fixed threshold heuristic. We chose to do this because the expected range of normally distributed residual values depends on the sample size which varied between models. Across all linear models, we excluded data points that produced an absolute standardised residual larger than the following cut-off threshold ($z$):

$$z = \Phi^{-1}\left(\frac{1}{2}\left(1 - 2^{-\frac{1}{n}}\right)\right) \tag{8}$$

Where, $\Phi^{-1}$ is the Probit function, and $n$ is the sample size. This threshold was chosen as it represents the bounds of a standard normal distribution that will contain all $n$ normally distributed data points of a random sample, 50% of the time. The value of $z$ is approximately 2.7 when $n = 100$ and 3.4 when $n = 1000$. After excluding outliers, Kolmogorov–Smirnov tests indicated that the residuals were normally distributed across all the linear mixed-effects models (across analyses, the proportion of excluded outliers ranged between 0 and 0.941%; see https://osf.io/cvm3r). Additionally, visual inspection of scatter plots showing residual versus predicted scores indicated no evidence of heteroscedasticity, non-linearity or overly influential datapoints (see https://osf.io/zpumq).

All *p*-values are reported as two-tailed statistics. Unless otherwise stated, we only report significant effects from the fMRI analyses that survive a Bonferroni correction for multiple comparisons across our 8 regions of interest.

## Supporting information

**S1 Text. A supplementary analysis demonstrating that, relative to interleaved training schedules, progressive training can bias some artificial neural networks to learn task representations that directly encode generalised relationships.**
(PDF)

**S2 Text. A report of significant effects that were identified in the representational similarity analyses but did not involve interactions with transitive distance and therefore did not directly relate to our a priori hypotheses.** As such, these results reflect changes in representational similarity that only vary by inferential accuracy, transitive slope, or experimental condition.
(PDF)

**S3 Text. A supplementary analysis of univariate and multivariate effects in the MPFC based on a finer parcellation of the cortex–the Schaefer et al [69]17-network, 400-region parcellation (see Methods).**
(PDF)

**S1 Table. Mean Spearman rank correlations between the number of correct responses to each inferred discrimination and the number of correct responses expected under the AND and OR models.** Rounded parentheses represent bootstrapped standard errors and square braces indicate bootstrapped 95% confidence intervals.
(PDF)

**S1 Fig. Univariate BOLD activity in the hippocampus did not significantly predict inference performance.** Panels **A** and **B** show activity in the left hippocampus. Panels **C** and **D** show activity in the right hippocampus. Bar charts display mean response amplitudes to all in-scanner discriminations split by trial type (premise vs inferred) and experimental condition (training method and session). Scatter plots display mean response amplitudes to all inference trials (both recent and remote) as a function of inference performance, split by training method (interleaved vs progressive). The only effect that reached statistical significance in these regions was detected in the right hippocampus. Here, a main effect of trial type indicated lower levels of BOLD activity on inference trails (panel C), yet this effect was not modulated by training method or inference performance. Individual data points indicate discrimination-specific BOLD estimates for each participant and error-bars indicate 95% confidence intervals.
(TIF)

**S2 Fig. In constructing the AND/OR models of human inference performance, memory for the premise discriminations was parametrised by computing a likelihood function of plausible values for the probability of correct retrieval ($L(r_p)$, where $p$ denotes a specific premise discrimination).** To do this we assume the probability of observing $k_p$ correct responses to the $n = 8$ test trials depends on a joint binomial process involving $r_p$ and, if retrieval is not successful, a random guess that yields a correct response with a probability of 0.5 (see Methods). Panel **A** presents the likelihood function for $r_p$ under different values of $k_p$. To approximate this distribution for the analysis, we randomly sampled values of $r_p$ using inverse transform sampling. This involved generating uniformly distributed random numbers in the range [0, 1] and selecting values of $r_p$ that returned cumulative likelihoods matching those random values (panel **B**).
(TIF)

**S3 Fig. Normalised group averages of each ROI in MNI space.** Each column relates to a different brain region and the blue-/orange- coloured overlays depict left-/right- hemisphere

ROIs (respectively). Overlay lightness represents ROI coverage across participants.
(TIF)

## Acknowledgments

We are grateful to Prof Neil Burgess and Prof Anil Seth for feedback on early drafts of the manuscript.

## Author Contributions

**Conceptualization:** Sam C. Berens, Chris M. Bird.

**Data curation:** Sam C. Berens.

**Formal analysis:** Sam C. Berens.

**Funding acquisition:** Chris M. Bird.

**Investigation:** Sam C. Berens.

**Methodology:** Sam C. Berens.

**Project administration:** Sam C. Berens, Chris M. Bird.

**Software:** Sam C. Berens.

**Supervision:** Chris M. Bird.

**Validation:** Sam C. Berens.

**Visualization:** Sam C. Berens.

**Writing – original draft:** Sam C. Berens, Chris M. Bird.

**Writing – review & editing:** Sam C. Berens, Chris M. Bird.

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
