## [Decision Letter · Decision Letter 0]

19 Apr 2022

Dear Dr Berens,

Thank you very much for submitting your manuscript "Hippocampal and medial prefrontal cortices encode structural task representations following progressive and interleaved training schedules" for consideration at PLOS Computational Biology.

As with all papers reviewed by the journal, your manuscript was reviewed by members of the editorial board and by several independent reviewers. In light of the reviews (below this email), we would like to invite the resubmission of a significantly-revised version that takes into account the reviewers' comments.

In particular, you should focus on improving cohesion between the computational, behavioural, and neural components of the manuscript, and make sure this work is placed in the context of existing literature on transitive inference.

We cannot make any decision about publication until we have seen the revised manuscript and your response to the reviewers' comments. Your revised manuscript is also likely to be sent to reviewers for further evaluation.

Sincerely,

Daniel Bush

Associate Editor

PLOS Computational Biology

Samuel Gershman

Deputy Editor

PLOS Computational Biology

Reviewer's Responses to Questions

**Comments to the Authors:**

Reviewer #1: This paper examines the effects of training schedule (progressive vs. interleaved) on the representations supporting transitive inference, using behavioral modeling, computational simulation, and fMRI. The authors hypothesize that interleaved training (pseudo-random presentation of premises) will lead to retrieval-based inference, such that individual premises are retrieved and integrated at the time of test; while progressive training (where each premise is well-learned before introducing another) will lead to the encoding of unified representations of the hierarchy during training. They also examine whether these different strategies are more or less prominent after short vs. long delays, hypothesizing that consolidation will increase the reliance on unified, "map-like" representations. The results showed clear advantages for progressive training over interleaved training in all conditions. Consistent positive symbolic distance effects (transitive slopes here) suggest a reliance on encoding-based inference rather than retrieval-based inference in both training conditions and both delays. The fMRI analysis generally supported the use of structural representations of the hierarchy rather than retrieval-based inference, though these effects were only apparent for more recently learned relationships.

The paper is generally well-written and clear. The theoretical background is mostly sufficient (with some important exceptions noted below) and the statistical analyses appear to be appropriate and well-documented.

The fMRI methods appear to be sound, although since this is not my area of expertise my review focuses on the behavioral results and computational modeling.

There are some notable strengths of the paper in terms of its originality and significance:

1. I agree that while both encoding- and retrieval-based mechanisms have been proposed as accounts of TI, there's a poor understanding of when people might rely on one or the other. So I found the overall research question to be interesting and worthwhile.

2. There are several existing computational models of encoding-based mechanisms for TI which can all capture the basic behavioral phenomena (like a positive symbolic distance effect). There's potential for model-based neuroimaging to make progress toward identifying which representations underlie behavior, particularly the kind of RSA approach used here.

3. Although both interleaved and progressive training have been commonly used in previous studies of TI, there are surprisingly few direct comparisons of different training schedules (particularly among humans). Aside from Markant (2020, 2021), the closest example would be Lazareva et al. (2020), who compared training conditions that were similar to the present paper in a sample of rhesus monkeys. They also found support for a unified representation regardless of training condition (or to be more precise, they didn't find evidence that simpler associative models could account for performance in either condition). The authors may want to incorporate this study in their background.

4. The authors have provided what appears to be a comprehensive repository of their code and materials on OSF.

I had some (relatively minor) concerns about the coverage of existing work on theories/models of TI (see point 1 below), and limitations with the task design which the authors may want to address in their discussion (see point 2).

I found the computational modeling components (both the MLP and the AND/OR comparison) to be the most problematic (see points 3 and 4 below). While both sections seem to be technically sound, they seemed to stand apart from the rest of the paper (particularly the MLP results, which seems to be an interesting modeling exercise but without any clear implications for how people perform TI). I expand on these points below.

# Main points

1. In some ways the paper does not sufficiently engage with the existing (large) literature on TI. Although I agree on the importance of the overall question of when/whether encoding- or retrieval-based mechanisms drive TI, the existing literature already suggests that encoding-based mechanisms dominate much of the time (indicated by positive transitive slopes seen in many studies). Besides the Banino et al. (2016) paper (which involved an associative inference task, not TI), I'm not aware of other studies that have shown negative transitive slopes as expected from a retrieval-based account (although there might be some evidence that slopes are more negative for individuals who are "unaware" of the hierarchy by the end of training, see e.g. Ryan et al., 2009).

More importantly, positive transitive slopes could be consistent with a number of different "encoding-based" mechanisms/models, including forms of reinforcement learning (e.g., Ciranka et al., 2021) and more map-like representations (like the particle filter model in Kumaran et al., 2016), and some other simpler associative learning models (see Vasconcelos, 2008). It's not altogether clear which of these are consistent with the "structural codes" described by the authors, or if it's possible to distinguish between these mechanisms on the basis of the present results. The authors discuss a couple of more recent proposals (the TEM and SR), but these other existing accounts are largely missing from the discussion.

As a side note, I was interested in the discussion of the successor representation as a potential alternative model (pg. 44). There seems to be some simulation work in the supplementary materials but this is only briefly mentioned in the discussion--it's unclear why this doesn't figure more prominently in the paper (there are also no corresponding methods to explain how they obtained the results in that figure).

2. There are some features of the task design which may have contributed to the reliance on an encoding-based strategy. The authors may want to consider how these limited their ability to detect retrieval-based inference and the generalizability of their results.

First, in the interleaved training condition there are many cases in which overlapping premises were experienced in successive trials, like the "chained" sequences in Markant (2021) which were associated with improved inference and greater explicit awareness of the hierarchy (the example in Figure 1C shows that this happened a fair amount of the time in the present task). This seems to run counter to the idea that "interleaved learning highlights the differences between items" (pg. 3), since it draws attention to shared elements across overlapping pairs. A training sequence without any overlap between successive pairs might be more likely to produce retrieval-based inference in line with the authors' expectations.

Second, when there are separable stimulus features people may also try to generalize based on those. The manuscript doesn't include many example of the textures, but looking at them on OSF there appear to be some dimensions that varied across items (e.g., color, size of the bricks). If people are given vague instructions about the task (i.e., to learn how the texture is related to reinforcement) then it would be reasonable that they compare across premises as they try to find a stimulus cue that predicts reinforcement. Granted that kind of strategy would not be successful in the task, but it's another factor that would lead to comparison across premises or attempts to form a more abstract representation of the hierarchy.

3. I struggled to understand the rationale for the comparison involving the AND and OR models (pg. 12). The AND model is a sensible (if simplistic) model of retrieval-based inference (a similar approach is taken in Ciranka et al., 2022). But I don't see the logic for the OR model, such that retrieval of a single premise leads to reactivation of the full representation of the hierarchy (or at least, the portion necessary to perform the inference at hand).

The authors state that these are not meant to be process models, but that is at odds with some other text that suggest mechanistic modeling, e.g.: "Our analyses show that the behavioural data are better accounted for by encoding-based generalisation mechanisms when directly compared with retrieval-based mechanisms" (pg. 4) (to me the heading "computational models of human inference" further gives the impression of process models of some type).

It would also seem that these models are simply re-describing the "transitive slope" results. It is helpful to see the individual variability depicted in Figure 5, but does this add anything beyond showing the distributions of transitive slopes? If so I think the authors need to be more explicit about the contribution here, including how this might improve statistical sensitivity (as they suggested in a response to an earlier review).

These shortcomings were especially notable since there are existing computational models of both retrieval- and encoding-based inference which might provide a stronger basis for detecting differences in strategies. In sum, I found this aspect of the modeling to ultimately be a distraction and to not add much beyond the behavioral results.

4. I found the MLP simulation results to be interesting and novel, but it was difficult to connect these results to the rest of the paper. The authors state that the models aren't intended to account for human performance or to adjudicate between encoding- and retrieval-based mechanisms, so it feels like quite a big detour in the middle of the paper with unclear implications.

It was also unclear to me whether increasing the number of hidden units was meant to capture a transition from an encoding-based strategy to a retrieval-based strategy (akin to the approach taken by Banino et al., 2016, where both encoding and retrieval-based inference were examined in the same connectionist framework). This seems not to be the case here--e.g. supplementary Figure 1 shows that with 6 hidden units the network still learns a value gradient but with poorer discrimination between the items (so there should still be a positive transitive slope). So in the end the MLP captures the difference between interleaving and progressive training to some extent, but can't speak to the other factors in question (including differences between encoding/retrieval and recent/remote conditions).

The MLP results might be more impactful if a stronger case was made about the importance of the interleaved vs. progressive training comparison. Most of the paper is framed around the goal of using these different training sequences to expose different mechanisms for inference, but the training conditions themselves are not inherently that interesting. The authors do allude to the reliance on interleaving in machine learning in the discussion—perhaps there are broader implications for work in that area, but this would seem to be somewhat out of place in the present paper.

# Minor points

- pg. 10: "Additionally, there was a significant 3-way interaction between training method, session and transitive distance, t(804) = 3.18, p = .002. This suggested that the effect of transitive distance was most consistent for remote discriminations in the progressive condition, t(804) = 3.07, p = .002, relative to all other conditions, t-values < 2.00 (see Figure 4B)." -- I don't follow this interpretation, what does it mean to be most "consistent"? From Figure 4B it looks like the effect of transitive distance is smallest in the progressive condition (and I'd be surprised if there was a difference between recent and remote for that condition).

- pg 11: For the GLM of RT, why is the DF so much larger than for the accuracy, when it is based on fewer observations?

# References

Ciranka, S., Linde-Domingo, J., Padezhki, I., Wicharz, C., Wu, C. M., & Spitzer, B. (2022). Asymmetric reinforcement learning facilitates human inference of transitive relations. Nature Human Behaviour. https://doi.org/10.1038/s41562-021-01263-w

Kumaran, D., Banino, A., Blundell, C., Hassabis, D., & Dayan, P. (2016). Computations underlying social hierarchy learning: Distinct neural mechanisms for updating and representing self-relevant information. Neuron, 92(5), 1135–1147.

Lazareva, O. F., Paxton Gazes, R., Elkins, Z., & Hampton, R. (2020). Associative models fail to characterize transitive inference performance in rhesus monkeys (Macaca mulatta). Learning & Behavior. https://doi.org/10.3758/s13420-020-00417-6

Markant, D. B. (2021). Chained study and the discovery of relational structure. Memory & Cognition. https://doi.org/10.3758/s13421-021-01201-1

Ryan, J. D., Moses, S. N., & Villate, C. (2009). Impaired relational organization of propositions, but intact transitive inference, in aging: Implications for understanding underlying neural integrity. Neuropsychologia, 47(2), 338–353.

Vasconcelos, M. (2008). Transitive inference in non-human animals: An empirical and theoretical analysis. Behavioural Processes, 78(3), 313–334. https://doi.org/10.1016/j.beproc.2008.02.017

Reviewer #2: Berens and Bird present a multifaceted study of transitive inference with a focus on different training schedules and consolidation. Through computational modelling, behaviour, and neuroimaging, they characterize how transitive inference is supported by interleaved and progressive learning schedules, hypothesizing that these schedules may encourage encoding- versus retrieval-based learning mechanisms. First, they find that simulations with MLPs show more consistent learning of inference relationships when trained with progressive schedules (notably, some interleaved schedules led to high inference performance but on average was at chance and highly variable). Second, they find strong a benefit in behaviour, notably for inference performance, for progressive schedules and demonstrate through a clever descriptive model analysis of premise pairs that participants were mostly likely engaged in encoding-based learning. Finally, the authors find that fMRI pattern similarity in hippocampus and mPFC is consistent with the transitive structure of the task. The authors argue that these findings collectively offer insight into how transitive inference learning can be optimized via progressive learning schedules and highlight potential neural mechanisms for building task-related representations although the link between behaviour and brain is murky.

The study is well conducted, the analyses are rigorous, and the interpretation of the findings is refreshingly honest. The comprehensive research approach that considers computation, behaviour, and the brain should be applauded and is a major strength of the current work. Also, the authors’ dedication and actual implementation of open science practices is noteworthy. It is also clear that each step of analysis has been carefully considered (e.g., generating nuisance variables that represent confounds introduced by pattern estimation is very clever). Despite this enthusiasm, I have to hesitate on the overall contribution of this work. As detailed below, I have important questions about the central neural findings and the overall cohesion of this study’s many components.

Major comments

- As noted above, I think the multifaceted approach of this study is a major strength. However, I feel that there is a notable lack of cohesion between the computational, behavioural, and neural components. For example, the MLP simulations (which were a surprise when first presented in the results section, more on that below) offer a computational starting point for considering the two training schedules and provide fairly straightforward predictions for human behaviour and neural measures, but how does this relate to encoding- vs. retrieval-based learning? And, the MLP findings seemingly had no conceptual bearing on the subsequent behavioural and neuroimaging analyses. This is largely a comment on framing and presentation; however, I think fixing this is necessary to fully evaluate the overall contribution of this work. A revised introduction that clearly walks through the overall approach and provides a motivation for the different components and how they are connected would greatly strengthen the manuscript.

- The key finding, which is highlighted in the title, is that structured task representations are found in hippocampus and mPFC. This is mostly supported by the RSA findings showing a negative relationship between transitive distance and representational similarity. But, the link between these neural signatures of task structure and transitive inference behaviour is weak. It is only those learners who perform poorly who drive this effect and it is stronger in the recent condition. The differences in how the mPFC and hippocampus effects relate to behaviour (e.g., that the hippocampus effect is strongest when behaviour did not suggest encoding-based learning) further clouds the interpretation. The result here is that the main finding seems to rely on a subset of the conditions tested and in a subset of participants who did not learn well. It is unclear what sort of neural representations supported successful learning. There may be some evidence that transitive structure is embedded in neural patterns, but it is not clear how well this is linked to behaviour (and in fact may be associated with poor learning or an unsuccessful strategy from those struggling with inferences). Given the clear evidence for a benefit in progressive training on inference performance, the neural findings based on a subset of participants in specific conditions feels like the two components are only loosely connected at best.

- Relatedly, the analysis pointing to neural representation of task structure (as depicted in Figure 7B&C) seems to rely on the existence of a linear relationship between transitive distance and pattern similarity. Conceptually, this makes perfect sense. But, a glance at the data depicted in Figure 7B&C suggests that this linear effect may be driven by a significantly higher similarity for delta 1 pairs relative to the other pairs with no appreciable difference between the delta 2-4 pairs. This coupled with the fact that the strongest effects here come from the recent condition suggest that rather than task structure, participants may instead be only representing the directly experienced premise pairs as more similar with no discernible structure among the other pairs. Is there evidence that these linear relationships truly represent that task structure consistent with the authors claims?

- Although I appreciate leveraging MLP simulations to provide computational predictions of the two training schedules, the motivation for this analysis is sparse. Why were MLPs used here, is there prior work that establishes this sort of model as particularly well-suited for predicting human-like transitive inference? Some of this is motivation/framing is provided in the Results section; but I think the authors’ approach would be much clearer if the MLP analysis was strongly motivated in the introduction.

- I appreciate the inclusion of the descriptive AND and OR models. It would be helpful, however, if connections could be drawn between the descriptive models and the MLP mechanisms. For example, are connection weights between the hidden and output layers in the MLP reflecting OR-type relationships? If possible, connecting these different analyses would strengthen the overall approach.

- The authors’ commitment to open science practices and a priori hypotheses is commendable. However, I wonder if their a priori choice of patterns based on anatomical and/or functional connectivity derived ROIs is limiting the type of neural evidence they can find. For example, the superior and inferior mPFC ROIs from the Schaefer et al. atlas divides this region into two units that don’t really correspond mPFC clusters/blobs identified in other studies of structure task representations. Also, in almost all of these ROIs (except for ERC) I wouldn’t expect that patterns from the entire extent of the ROI would hold relevant information for the task. A searchlight analysis that considered more localized patterns within the ROIs may be more sensitive to the type of behaviourally related task structure the authors are interested in. I understand that this goes off script and is a clear “exploratory” analysis, but it may provide further insight into the neural mechanisms underlying the clear advantage for progressive training.

- I can infer some of the history in this work from the included response to prior reviewers. And, it is clear that updated analyses may have changed some of the initial findings. But, in its current form, I found that the ROI selection was poorly motivated and the omission of some of these ROIs from the results was confusing. For example, what happened with ERC? I realize that the supplement includes more details about the ROIs and the main text is reserved for the central significant results. But a more thorough description and/or figures that include the results from all ROIs would be help clarify the findings.

Minor comments:

- Overall, I ran into many typos or missing words. In one place, Figure 9 was referred to but I think should be Figure 8. Apologies that I did not note all of these here, but a careful read and edit of the manuscript is needed.

- Typo: Second paragraph of Results: “They were not indented...”, should be “intended”.

- The first subheading of the Results section doesn’t quite reflect the following text as both progressive and interleaved training results are presented. Perhaps the intention here was to refer to the better performance from progressive schedules; if so, the subtitle should be edited to reflect this.

- It is not clear what the data points represent in each figure. For example, in Figure 5, each connected pair of data points is presumably relating to a single subject. Figure 4C and D must be representing trial-by-trial RTs. This should be explained in the figure legend.

- A complementary paper to the Zhou et al work [43] is by Heffernan et al. 2021 who demonstrated that progressive schedules can be more advantageous in human and model for learning category exceptions. Could be worth including in the current discussion.

- I typically consider supplements an open range where authors can include whatever they want. But I was completely befuddled by the inclusion of the successor representation description in Supplemental Figure 4. It is perhaps an interesting point that SRs can represent transitive inference, but this seems like a departure from the current study.

**Have the authors made all data and (if applicable) computational code underlying the findings in their manuscript fully available?**

Reviewer #1: Yes

Reviewer #2: Yes

PLOS authors have the option to publish the peer review history of their article (what does this mean?). If published, this will include your full peer review and any attached files.

Reviewer #1: No

Reviewer #2: No
---

## [Decision Letter · Decision Letter 1]

25 Aug 2022

Dear Dr Berens,

Thank you very much for submitting your manuscript "Hippocampal and medial prefrontal cortices encode structural task representations following progressive and interleaved training schedules" for consideration at PLOS Computational Biology. As with all papers reviewed by the journal, your manuscript was reviewed by members of the editorial board and by several independent reviewers. The reviewers appreciated the attention to an important topic. Based on the reviews, we are likely to accept this manuscript for publication, providing that you modify the manuscript according to the review recommendations. In particular, please address the final outstanding concerns of Reviewer 1.

Sincerely,

Daniel Bush

Academic Editor

PLOS Computational Biology

Samuel Gershman

Section Editor

PLOS Computational Biology

[LINK]

Reviewer's Responses to Questions

**Comments to the Authors:**

Reviewer #1: The authors have made extensive revisions that appropriately address my previous concerns. I find the new version to be much more streamlined and easier to follow. I appreciate the authors' thoughtful and thorough responses to the previous reviews.

I have a couple of minor remaining points of clarification that the authors may want to address:

1. Pg. 3, bottom: I still think the description of the effects of transitive distance on accuracy could be improved. I appreciate the authors' point that there can be a positive effect in the progressive/remote condition despite being near ceiling for overall accuracy. Is there not also a difference in the effects among interleaved learners, with a stronger effect of transitive distance in the recent condition compared to the remote condition? It may be helpful here to state the effects of transitive distance in each condition separately, or to say more directly that the effect was only significant (rather than "most strongly correlated") in the progressive-remote condition ("t-values < 2.00" suggests this but this is imprecise and easy to overlook, especially when the figure seems to suggest a difference between the interleaved conditions). I press this point because there are differences in the corresponding effects in the subsequent fMRI results, so it seems important to establish what the overall behavioral effects are in each condition.

2. Pg. 22: In the AND/OR model descriptions, what is R_i? Is it supposed to be A_i, or are these different?

Reviewer #2: The revised manuscript and responses from Behrens and Bird offers a comprehensive answer to my initial concerns. Indeed, the authors argument detailing the extent of their effects clarified my original (misguided) concern. Moreover, the control analyses investigating the linear relationship between transitive distance and pattern similarity and the additional look at a more fine-grained parcellation of mPFC provide compelling findings in support of the central claims. Finally, the edits to the revised manuscript (e.g., excluding the MLP analysis and SR simulations) provide a more concise presentation. I appreciate the authors' consideration of the reviewer comments and their detailed reply and control analyses. Overall, this work offers a valuable contribution to the field and I look forward to seeing it published.

**Have the authors made all data and (if applicable) computational code underlying the findings in their manuscript fully available?**

Reviewer #1: Yes

Reviewer #2: Yes

PLOS authors have the option to publish the peer review history of their article (what does this mean?). If published, this will include your full peer review and any attached files.

Reviewer #1: No

Reviewer #2: No

Figure Files:

Data Requirements:

Reproducibility:

References:

---

## [Decision Letter · Decision Letter 2]

13 Sep 2022

Dear Dr Berens,

We are pleased to inform you that your manuscript 'Hippocampal and medial prefrontal cortices encode structural task representations following progressive and interleaved training schedules' has been provisionally accepted for publication in PLOS Computational Biology.

Best regards,

Daniel Bush

Academic Editor

PLOS Computational Biology

Samuel Gershman

Section Editor

PLOS Computational Biology

Reviewer's Responses to Questions

**Comments to the Authors:**

Reviewer #1: The authors have appropriately addressed my comments from the previous revision.

**Have the authors made all data and (if applicable) computational code underlying the findings in their manuscript fully available?**

Reviewer #1: Yes

PLOS authors have the option to publish the peer review history of their article (what does this mean?). If published, this will include your full peer review and any attached files.

Reviewer #1: No

---

## [Editor Report · Acceptance letter]

12 Oct 2022

PCOMPBIOL-D-22-00347R2 

Hippocampal and medial prefrontal cortices encode structural task representations following progressive and interleaved training schedules

Dear Dr Berens,

I am pleased to inform you that your manuscript has been formally accepted for publication in PLOS Computational Biology. Your manuscript is now with our production department and you will be notified of the publication date in due course.

With kind regards,

Anita Estes
